# DOCK1 insufficiency disrupts trophoblast function and pregnancy outcomes via DUSP4-ERK pathway

Yichi Xu[1,2,3,*] , Xiaorui Liu[1,2,3,*], Weihong Zeng[1,2,3], Yueyue Zhu[1,2,3], Junpeng Dong[1,2,3], Fan Wu[1,2,3], Cailian Chen[4,5], Surendra Sharma[6], Yi Lin[7] 

Abnormal trophoblast function is associated with diseases such as recurrent spontaneous abortion, pre-eclampsia, and preterm birth, and endangers maternal and fetal health. However, the underlying regulatory mechanisms remain unclear. In this study, we found DOCK1 expression is decreased in the placental villi of patients with recurrent spontaneous abortion, and that its expression determined the invasive properties of extravillous trophoblasts (EVTs), highlighting a previously unknown role of DOCK1 in regulating EVT function. Furthermore, DOCK1 deficiency disturbed the ubiquitinated degradation of DUSP4, leading to its accumulation. This caused inactivation of the ERK signaling pathway, resulting in inadequate EVT migration and invasion. DOCK1 was implicated in regulating the ubiquitin levels of DUSP4, possibly by modulating the E3 ligase enzyme HUWE1. The results of our in vivo experiments confirmed that the DOCK1 inhibitor TBOPP caused miscarriage in mice by inactivating the DUSP4/ERK pathway. Collectively, our results revealed the crucial role of DOCK1 in the regulation of EVT function via the DUSP4-ERK pathway and a basis for the development of novel treatments for adverse pregnancy outcomes caused by trophoblast dysfunction.

## Introduction

The placenta, which serves as an interface between the fetus and the mother, is essential for normal fetal development and fetal and maternal health during mammalian gestation. The major functions of the placenta are carried out by the trophoblast cells, which originate from the outer layer of the blastocyst, the trophectoderm (TE) (Turco & Moffett, 2019). After embryo implantation, the trophectoderm undergoes a transformation, generating the cytotrophoblast (CTB), which is considered a progenitor cell population. The CTB has the ability to differentiate into two distinct lineages: the extravillous trophoblast (EVT) and the syncytiotrophoblast (Velicky et al, 2016; Mischler et al, 2021). The EVT is crucial for placental fixation, dilation of the uterine spiral arteries, and protection against the maternal immune response towards the placenta and fetal tissues (Moffett & Loke, 2006; Tilburgs et al, 2015). The EVT invades the maternal decidua to some extent; this is essential for normal placentation and successful pregnancy (Li et al, 2021).

The dysregulation of EVT functions is closely linked to pregnancy complications, including preeclampsia, intrauterine growth restriction, and retained placenta. These complications collectively contribute to over 300,000 maternal deaths and ~15 million preterm births worldwide annually (Park et al, 2022). Inadequate EVT invasion and impaired spiral artery remodeling have been linked to preeclampsia—a hypertensive disorder that affects the mother and the fetus (Staff et al, 2022). Insufficient blood supply to the placenta because of impaired spiral artery remodeling can also cause fetal growth restriction, where the fetus fails to reach its full growth potential (Staff et al, 2022). Growing evidence uncovered that defects in trophoblast proliferation, shallow invasion, and excessive trophoblast apoptosis play significant roles in the occurrence of recurrent spontaneous abortion (RSA) (Lamptey et al, 2021). A study in Japan revealed consistent live birth rates in women with RSA from 1994–2010 (67.8%) to 2011–2018 (63.8%) (Morita et al, 2019), indicating no improvement in outcomes since the mid-1990s. Furthermore, nearly half of RSA cases have an unknown cause and are medically termed as unexplained RSA. There is limited knowledge regarding the specific mechanisms involved in these conditions. Understanding the complex processes of trophoblast proliferation, differentiation, invasion, and spiral artery remodeling is essential for identifying potential causes and developing interventions to prevent or treat infertility and adverse pregnancy outcomes.

The mammalian DOCK (Dedicator of Cytokinesis) family comprises 11 proteins (DOCK1–DOCK11) which perform various indispensable

[1]The International Peace Maternity and Child Health Hospital, School of Medicine, Shanghai Jiao Tong University, Shanghai, China   [2]Shanghai Key Laboratory of Embryo Original Diseases, Shanghai, China   [3]Institute of Birth Defects and Rare Diseases, School of Medicine, Shanghai Jiao Tong University, Shanghai, China   [4]Department of Automation, School of Electronic Information and Electrical Engineering, Shanghai Jiao Tong University, Shanghai, China   [5]Key Laboratory of System Control and Information Processing, Ministry of Education of China, Shanghai, China   [6]Department of Pediatrics, Women and Infants Hospital of Rhode Island, Warren Alpert Medical School of Brown University, Providence, RI, USA   [7]Center of Reproductive Medicine, Department of Obstetrics and Gynecology, Shanghai Sixth People's Hospital Affiliated to Shanghai Jiao Tong University School of Medicine, Shanghai, China

Correspondence: yilinonline@126.com
*Yichi Xu and Xiaorui Liu contributed equally to this work

  

cellular functions such as regulation of the actin cytoskeleton, cell adhesion, and migration (Gadea & Blangy, 2014). DOCK1 (dedicator of cytokinesis 1), also known as DOCK180, is a guanine nucleotide exchange factor that triggers Rac1 activation and promotes cell migration, invasion, proliferation, and survival (Rossman et al, 2005). DOCK1 deficiency leads to pregnancy failure by modulating decidualization and angiogenesis in the mouse uterus (Mohan et al, 2018); however, the exact mechanism remains unclear. DOCK1 has emerged as a key driver of cancer metastasis in multiple cancer types (Feng et al, 2011; Feng et al, 2012; Laurin et al, 2013; Tomino et al, 2018). Owing to the broad similarities between tumor and EVT cells in cell proliferation, migration, and invasion, we hypothesized that DOCK1 is likely involved in regulating trophoblast function.

DUSP4, also known as MKP-2, is a member of the dual-specificity phosphatases (DUSPs) family that can dephosphorylate phospho-tyrosine and phosphor-threonine residues in MAP kinases, including p38 MAPK, extracellular signaling-associated kinase (ERK), and JNK/stress-activated protein kinase (Wu, 2007). DUSP4 suppresses cell growth by dephosphorylation of ERK kinases and reduces cell invasiveness by inhibiting phosphorylation in the MAPK (ERK, p38, and JNK) signaling pathway (Carlos et al, 2013; Mazumdar et al, 2016).

In this study, we used both in vivo and in vitro models to examine the role of DOCK1 in trophoblast function and pregnancy outcomes. Our investigation focused on the expression of DOCK1 in the placental villi of patients with RSA, and we found that its expression status determined the invasive properties of EVTs, highlighting a previously unknown role of DOCK1 in regulating EVT function during early pregnancy. Furthermore, our study showed that the significant role of DOCK1 was associated with the involvement of the ERK signaling pathway and DUSP4 as intermediaries. DOCK1 was implicated in regulating the ubiquitin levels of DUSP4, possibly by modulating the E3 ligase enzyme HUWE1. In animal models, we identified the critical role of DOCK1 in facilitating successful pregnancy, as evidenced by using the DOCK1 inhibitor TBOPP. Insufficient trophoblast function can result in placental dysfunction and the onset of various pregnancy complications. These complications have profound effects on both maternal health and fetal development, potentially influencing the health of the offspring throughout their life. Studying the molecular mechanisms of DOCK1 in trophoblast dysfunction could provide a basis for the discovery of therapeutic strategies for recurrent miscarriage, preterm labor, preeclampsia, and other pregnancy complications.

# Results

## DOCK1 expression is down-regulated in RSA

Eleven protein subtypes of the mammalian DOCK family, namely DOCK1 to DOCK11, are involved in various essential cellular functions, including actin cytoskeleton reorganization, cell migration, and adhesion (Gadea & Blangy, 2014). To determine the role of DOCK family subtypes in modulating EVT functions and the onset of EVT dysfunction-related diseases such as RSA, we studied DOCK family gene expression in placental villi from patients with RSA by using

real-time quantitative polymerase chain reaction (qRT–PCR). DOCK1 and DOCK4 mRNA expression was significantly lower in patients with RSA than in healthy control (HCs). There were no significant differences in the expression of other DOCK family genes (Fig 1A).

Considering the clear difference in DOCK1 expression in HCs and RSA, and its essential functions during pregnancy establishment, involving decidualization and angiogenesis in the mice uterus (Mohan et al, 2018), we focused on the role of DOCK1 in RSA pathogenesis. We used Western blot analysis to determine the protein levels of DOCK1 in placental villi tissues obtained from HCs and patients with RSA. DOCK1 protein expression was decreased in the villus of patients with RSA (Fig 1B). Subsequently, we studied in situ DOCK1 protein expression in placental tissues. EVTs were identified using an HLA-G antibody. We observed weak DOCK1 immunoreactivity in the EVTs of patients with RSA, and strong fluorescence signals in the EVTs of HCs (Fig 1C). To identify trophoblasts, an anti-cytokeratin 7 (CK7) antibody was employed, revealing a faint DOCK1 immunoreactivity in the syncytiotrophoblasts and CTBs of RSA patients (Fig 1D). Thus, DOCK1 mRNA and protein were down-regulated in the placental villi tissues of patients with RSA, indicating that DOCK1 may play a pivotal role in the pathogenesis of adverse pregnancy outcomes.

## DOCK1 is required for trophoblast proliferation and cell cycle progression

Functional studies on human implantation have predominantly used cultured human trophoblasts as a means to replicate and study EVT proliferation and invasion in vitro. The HTR-8/SVneo cell line is the most widely used cell model for EVT cell biology and was established from human first-trimester EVT cells (Graham et al, 1993). To determine the function of DOCK1 in EVTs, we used CRISPR/Cas9 technique and exogenous plasmid introduction to deplete and overexpress DOCK1, respectively, in HTR-8 cells, and confirmed the efficiency of knockout and overexpression (Fig S1A and D). Studies using JAR cells have also provided insights on trophoblast function. We therefore knocked down or overexpressed DOCK1 in the JAR trophoblast cell line, and confirmed the efficiency of knockdown or overexpression (Fig S1B, C, and E). To determine the functional role of DOCK1 in EVTs, we evaluated the proliferation ability of HTR-8 cells with DOCK1 deletion or overexpression. Our results showed that the depletion of DOCK1 reduced cell proliferation in HTR-8 cells, as evidenced by CCK-8 (Fig 2A), cell colony formation (Fig 2B), and Edu assays (Fig 2C). Conversely, DOCK1 overexpression promoted cell proliferation (Fig 2A–C). These findings were also observed in JAR cells (Fig S2A–C). These data highlight the indispensable role of DOCK1 in trophoblast proliferation.

Further insights into the mechanism underlying the observed proliferation deficits were garnered through cell cycle analysis. Notably, inhibition of DOCK1 or treatment with 10 $\mu$M TBOPP for 24 h in HTR-8 cells resulted in a significant arrest of cells in the G0/G1 phase, with a concomitant reduction in S phase cell population (Fig 2D and E). A similar trend was observed in JAR cells (Fig S2D and E). The relationship between cell proliferation and the cell cycle is intrinsic. Any interruption or arrest, such as what we observed in the G0/G1 phase because of DOCK1 inhibition, can have effect on the overall proliferative capacity of the cell. The inhibition of DOCK1

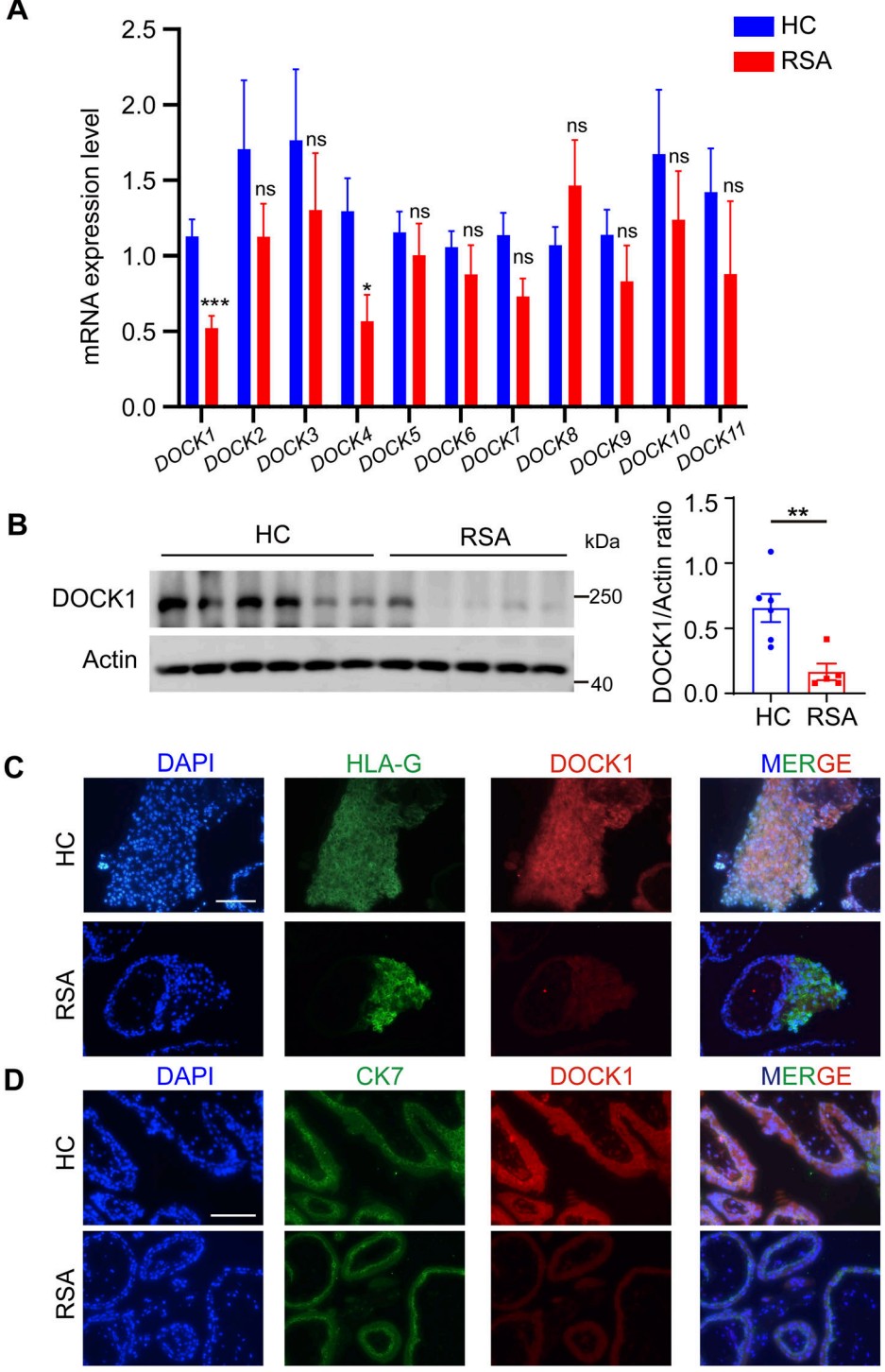

**Figure 1. DOCK1 is down-regulated in placental tissues after unexplained recurrent spontaneous abortion (RSA).**
**(A)** Comparison of relative mRNA expression levels of the *DOCK* family in villous tissue from patients with RSA (n = 12) and HCs (n = 9). **(B)** DOCK1 protein content in HCs (n = 6) and RSA (n = 5) determined using Western blotting (Left). Comparison of DOCK1 protein content in HCs and RSA (Right). **(C)** Representative immunohistochemical staining of DOCK1 in extravillous trophoblasts. Scale bar: 200 *μm*. **(D)** Representative immunohistochemical staining of DOCK1 in cytotrophoblasts and syncytiotrophoblasts. Scale bar: 200 *μm*. **(A, B)** Data information: in (A, B), analysis of differences was determined with unpaired two-tailed *t* test. *$P < 0.05$, **$P < 0.01$. ns, not significant.

exerts a multifaceted impact. Not only does it attenuate the proliferation capacity of trophoblast cells by directly impeding their growth, but it also disrupts the natural progression of the cell cycle, especially arresting it in the G0/G1 phase. These results reinforce the notion that DOCK1 is essential not only for cell proliferation but also for playing a crucial role in cell cycle progression.

## DOCK1 regulates EVT migration and invasion

In addition to the proliferative phenotype, we aimed to study EVT migration and invasion, which is important for the establishment of normal pregnancy. We therefore conducted transwell experiments to measure the migration and invasiveness of HTR-8 cells with or

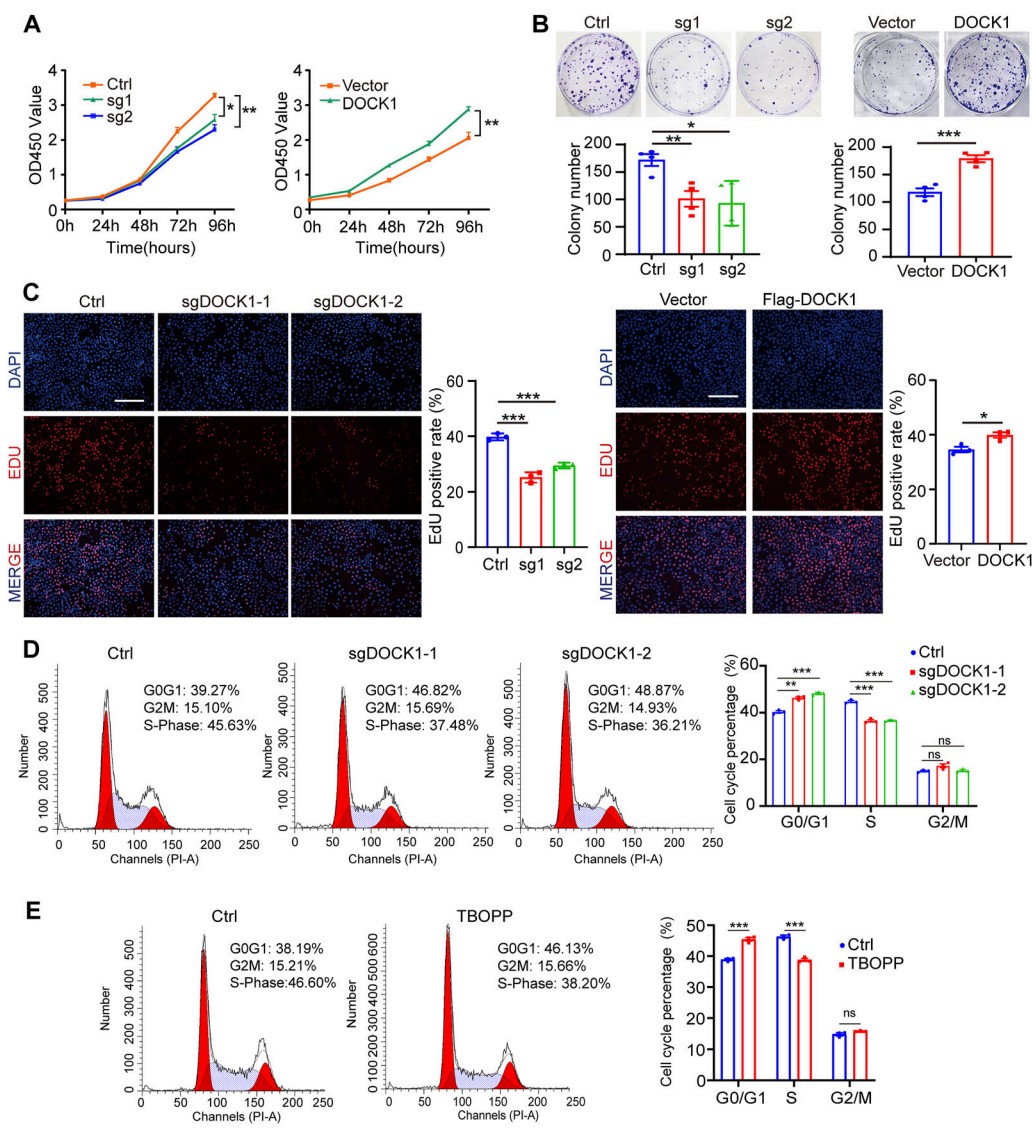

**Figure 2. DOCK1 modulates cell proliferation and cell cycle progression.**
**(A)** CCK-8 assay results showing the viability of HTR-8 cells after DOCK1 repression or overexpression. **(B)** Cell colony formation assays to determine the clonogenicity of HTR-8 cells after DOCK1 repression or overexpression (top), and cell quantification (bottom). **(C)** Edu assay results showing the proliferation of HTR-8 cells after DOCK1 repression or overexpression (left). The Edu-positive rate was quantified (right). Scale bar: 200 $\mu$m. **(D)** Flow cytometry analysis showing the distribution of HTR-8 cells among G0/G1-, S-, and G2/M-phases after DOCK1 knockout (left), and percentage analysis of cells in each phase relative to the total (right). **(E)** After HTR-8 cells were treated with DOCK1 inhibitor 10 $\mu$M TBOPP for 24 h, cell cycle distribution was analyzed by flow cytometry (left). The respective percentages of cells in each phase are presented (right). In (A, C, D, E), each group, n = 3 biological replicates (*$P < 0.05$; **$P < 0.01$; ***$P < 0.001$; unpaired two-tailed $t$ test). In (B), each group, n = 4 biological replicates (*$P < 0.05$; **$P < 0.01$; ***$P < 0.001$; unpaired two-tailed $t$ test).

without the addition of matrigel. Silencing DOCK1 expression resulted in decreased migration and invasion ability, and ectopic expression of DOCK1 had the opposite effects on HTR-8, as shown by transwell assays (Fig 3A). In addition, we used wound healing assays to evaluate cell migration, and observed that the down-regulation of DOCK1 inhibited cell migration, and the up-regulation of DOCK1 promoted it (Fig 3B). Similar effects were observed in JAR cells, where DOCK1 down-regulation suppressed cell motility, and overexpression exerted the opposite effects (Fig 3C and D).

When HTR-8 cells were treated with 10 $\mu$M TBOPP for 24 h, migration and invasion significantly decreased (Fig 3E and F). These

observations highlight the critical role of DOCK1 in supporting cell motility.

## DOCK1 expression promotes nuclear accumulation of $\beta$-catenin and triggers epithelial–mesenchymal transition (EMT)

We then investigated the potential mechanism by which DOCK1 affects cell motility. Specifically, we examined the expression of nuclear $\beta$-catenin in HTR-8 and JAR cells after DOCK1 inhibition by using Western blot and immunofluorescence staining. Western blot assays showed that reduced DOCK1 impaired the nuclear

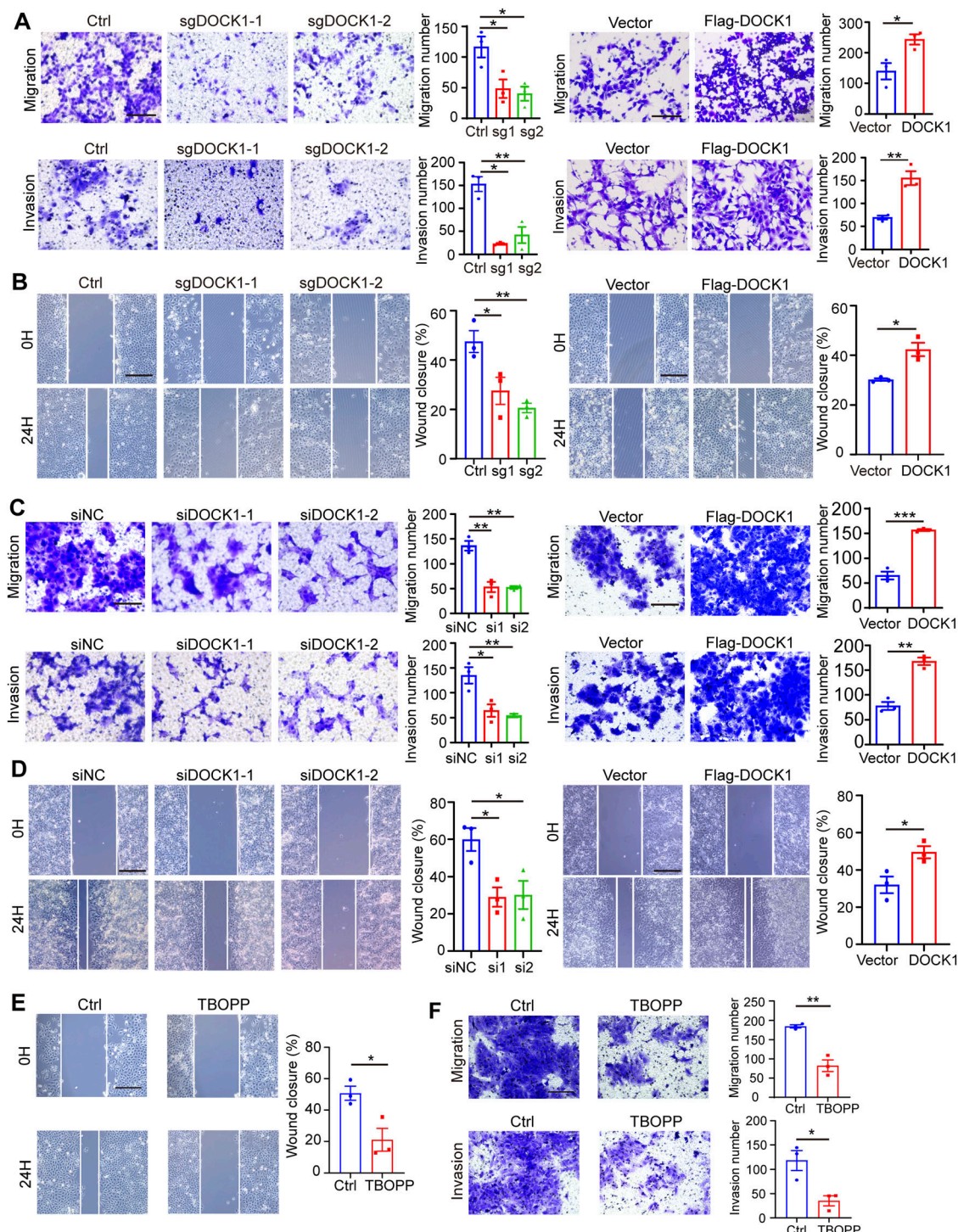

**Figure 3. DOCK1 promotes the migration and invasion.**
**(A)** Transwell assay results showing the migration and invasion of HTR-8 cells DOCK1 depletion or overexpression (left). The number of migrating and invading cells was quantified (right). Scale bar: 100 $\mu$m. **(B)** Wound healing assay results showing the migration of HTR-8 cells after DOCK1 repression or overexpression (left). The rate of wound closure was quantified (right). Scale bar: 200 $\mu$m. **(C)** Transwell assay results showing JAR cell migration and invasion after DOCK1 inhibition or overexpression (left) and quantification of migrating or invading cells (right). Scale bar: 100 $\mu$m. **(D)** Wound healing assays results showing JAR cell migration after DOCK1 repression or overexpression (left) and quantification of rate of wound closure (right). Scale bar: 200 $\mu$m. **(E)** Wound healing assay results showing the migration ability of HTR-8 cells after treatment with 10 $\mu$M TBOPP (left) and quantification of rate of wound closure (right). Scale bar: 200 $\mu$m. **(F)** Transwell assays showing the migration and invasion ability of HTR-8 cells after TBOPP treatment (left). The number of migrating and invading cells was calculated (right). Scale bar: 100 $\mu$m. In (A, B, C, D, E, F), each group, n = 3 biological replicates (*$P < 0.05$; **$P < 0.01$; ***$P < 0.001$; unpaired two-tailed t test).

expression of β-catenin (Fig 4A and B). This result was confirmed by immunofluorescence staining, which demonstrated disrupted nuclear localization of β-catenin in DOCK1 knockout cells (Fig 4C).

During early gestation, EVTs are transformed into invasive cells through a partial EMT. This transition involves the replacement of epithelial-like with mesenchymal-like morphology (Zhou et al, 1997). Inhibition of DOCK1 caused significant changes in cell morphology with inhibition of EMT. DOCK1-knockout HTR-8 cells transitioned from a spindle-like mesenchymal shape to a round-like epithelial shape (Fig 4D). In our study, DOCK1-knockout HTR-8 cells exhibited reduced levels of the mesenchymal-like molecules N-cadherin, vimentin, and snail (Fig 4E). In contrast, DOCK1 over-expression had the opposite effect (Fig 4E). In line with these observations, the protein levels of the migration markers MMP2 and MMP9 decreased in DOCK1-depleted HTR-8 cells, and increased in cells with DOCK1 overexpression (Fig 4E). In JAR cell, suppression of DOCK1 led to decreased levels of vimentin, snail, MMP2, and MMP9, whereas it up-regulated the expression of the epithelial-like molecule E-cadherin (Fig 4F). In contrast, overexpressing DOCK1 in JAR cells had the reverse effect (Fig 4F). These findings indicate that DOCK1 is required to maintain the EMT state and invasiveness of EVT.

## DOCK1 modulates trophoblast motility through the ERK signaling pathway

In the next phase of our study, we aimed to identify the specific signaling pathways and molecular mechanisms by which DOCK1 acts on EVTs. We conducted differential transcriptome expression profiling analyses between DOCK1 knockout HTR-8 cells and control cells. In the DOCK1-knockout cells, 264 genes were highly expressed, whereas 399 genes were down-regulated (Fig S3A). Based on Gene Ontology (GO) analysis, we listed the top 10 most prominent terms associated with DOCK1 expression. DOCK1 was associated with numerous biological processes (BP), especially cell and tissue migration (Fig 5A), consistent with our experimental results in trophoblasts. Furthermore, according to the Kyoto Encyclopedia of Genes and Genomes pathway analysis, DOCK1 knockdown resulted in the most significant enrichment of differentially expressed genes (DEGs) in the MAPK signaling pathway, as shown in Fig 5B. Thus, we studied the phosphorylation levels of ERK at Tyr202/204, a key component of the MAPK signaling pathway, in DOCK1 knockout cells. As expected, DOCK1-depleted cells displayed deactivated ERK signaling. In contrast, overexpression of DOCK1 contributed to activating the ERK signaling pathway (Fig 5C). Similar results were found in JAR cells (Fig S3B).

Based on the RNA sequencing analyses and our experimental results, we then determined whether the reduction of cell mobility in DOCK1 knockout cells was related to the inactivation of the ERK pathway. Epidermal growth factor (EGF), an essential stimulator of the ERK signaling pathway, was added to DOCK1 knockout cells. We selected 10 ng/ml EGF to treat cells for 30 min based on the increased phosphorylation of ERK in HTR-8 cells (Fig S3C). EGF treatment activated ERK phosphorylation in DOCK1 knockout cells, and the loss of the migration marker proteins MMP2, MMP9, vimentin, and snail was rescued by EGF treatment (Fig 5D). Furthermore, ERK signaling activation rescued the inhibition of wound

healing ability, migration, and invasion in DOCK1 knockout cells (Fig 5E and F). These data indicate that the mechanism by which DOCK1 induces trophoblast motility involves the activation of the ERK signaling pathway.

## DOCK1 modulates cell migration and invasion via the DUSP4-ERK signal pathway

DUSP4 is a pivotal molecular in the MAPK signaling pathway, especially in suppressing cell invasion by inactivating ERK phosphorylation (Hijiya et al, 2016). We therefore hypothesized that DOCK1 may regulate the ERK pathway by targeting DUSP4. We evaluated the mRNA and protein expression of DUSP4 in DOCK1 knockout cells by using qRT–PCR and Western blotting, and observed that DOCK1 knockout caused less than a twofold increase in DUSP4 mRNA expression (Fig 6A). However, the modulation of DUSP4 protein appeared more pronounced, with a fourfold increase, suggesting that DOCK1 regulated DUSP4 expression partly through translational mechanisms or posttranslational modifications (Fig 6B). Furthermore, the depletion of DUSP4 did not affect DOCK1 expression, providing further evidence that DUSP4 is a downstream molecule of DOCK1 (Fig S3D) in the DOCK1-ERK signaling pathway.

To validate the functional significance of DUSP4 in cell migration and invasion, we performed wound healing and transwell assays. DUSP4 knockdown successfully rescued impaired migration and invasion in HTR-8 cells (Fig 6C and D). In the absence of DOCK1, silencing DUSP4 with siRNA led to compensatory activation of the ERK pathway, resulting in increased ERK phosphorylation and elevated protein levels of the invasion markers MMP2, MMP9, vimentin, and snail (Fig 6E). These results suggest a critical role for the DUSP4-ERK signal pathway in mediating DOCK1-regulated invasiveness of EVTs.

## DOCK1 disrupts DUSP4 protein stability

We determined the expression of DUSP4 protein in DOCK1 knockout cells and found that DOCK1 depletion significantly increased DUSP4 protein levels (Fig 6B). Conversely, DOCK1 overexpression markedly reduced DUSP4 protein levels in trophoblasts, suggesting negative DUSP4 modulation by DOCK1 (Fig 7A). To elucidate the potential interaction between DOCK1 and DUSP4, we performed co-immunoprecipitation assays using lysates from HEK293T cells expressing exogenous Flag-DOCK1 and Myc-DUSP4, confirming their interaction (Fig 7B). Furthermore, we observed partial co-localization of endogenous DOCK1 with DUSP4 in the cytoplasm, indicating a possible physiological interaction (Fig 7C).

To further characterize the interaction between DOCK1 and DUSP4, we treated trophoblasts with a protein synthesis inhibitor, cycloheximide (CHX), and found that DOCK1 depletion enhanced the stability of the DUSP4 protein, while DOCK1 overexpression impaired its stability (Fig 7D and E). These results indicate that DOCK1 modulates DUSP4 protein at the posttranslational level. In eukaryotic cells, there are mainly two degradation systems: the autolysosome–lysosomal and ubiquitin–proteasomal systems (Cohen-Kaplan et al, 2016). The ubiquitin–proteasome system primarily degrades short half-life proteins, whereas the cellular lysosomal system is responsible for degrading long half-life

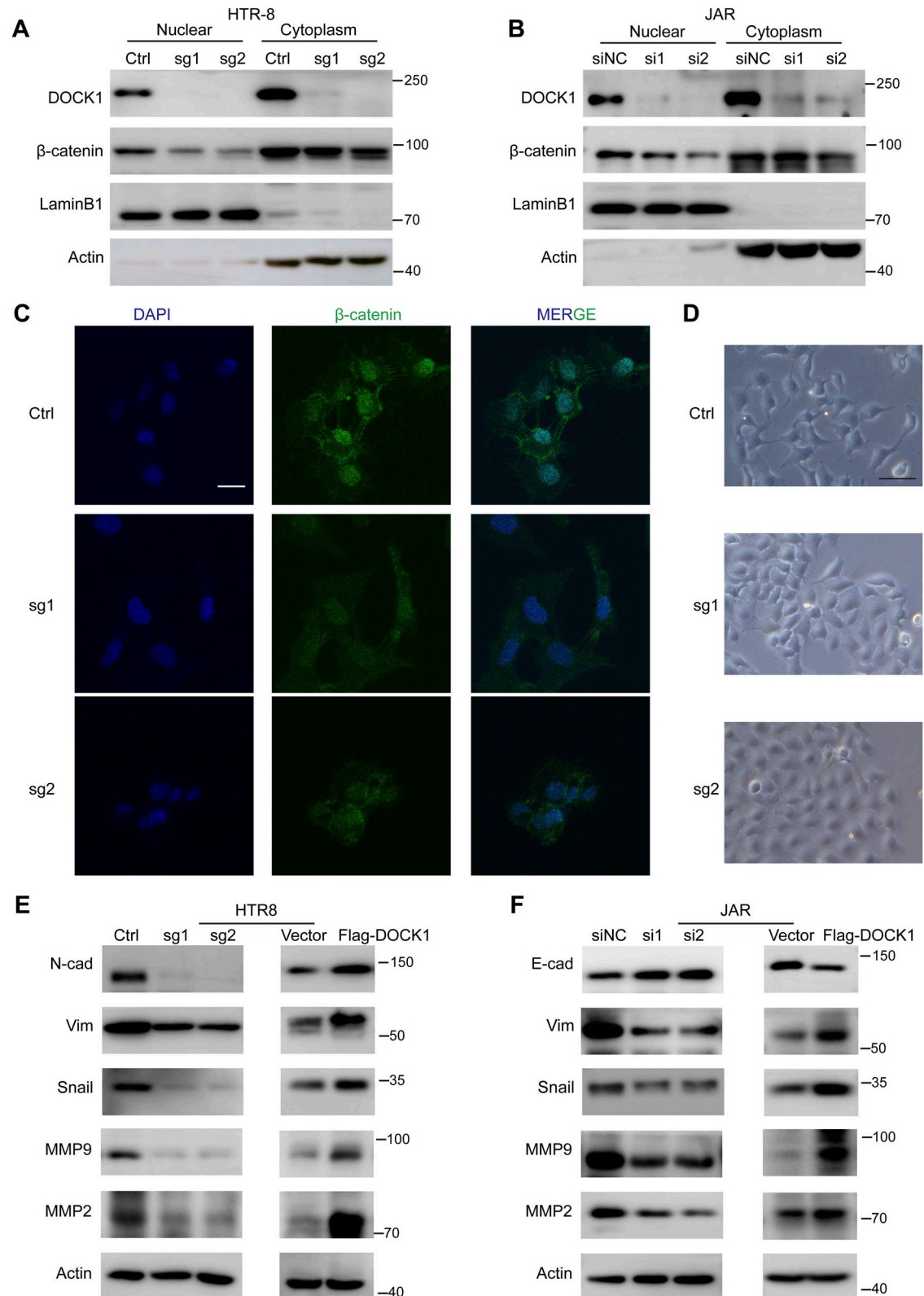

**Figure 4. DOCK1 promotes the epithelial–mesenchymal transition (EMT) process.**
**(A)** β-catenin protein levels in different subcellar locations of HTR-8 cells depleted of DOCK1 analyzed by Western blot. **(B)** Western blot analysis of β-catenin protein levels in different subcellar locations of JAR cells transfected with siDOCK1 or siNC. **(C)** Representative images of β-catenin immunofluorescence staining in DOCK1 knockout HTR-8 cells. Scale bar: 25 μm. **(D)** Morphological changes in HTR-8 cells after DOCK1 knockout. Scale bar: 50 μm. **(E)** Western blot results showing EMT or

proteins (Shah & Di Napoli, 2007; Rajawat et al, 2009). According to our results, DUSP4 is a short half-life protein, which is consistent with previous reports (Torres et al, 2003; Huang & Tan, 2012). Thus, the degradation of the DUSP4 protein likely occurs through the ubiquitin–proteasome system.

We verified this hypothesis by examining DUSP4 protein levels in cells treated with 50 μg/ml CHX for 6 h. DUSP4 protein was reduced with CHX treatment, and the addition of 20 μM proteasome inhibitor MG132 for 6 h reversed the decrease in DUSP4 protein expression (Fig 7F). However, treatment with 10 μM of the lysosomal inhibitor chloroquine (CQ) or 1 mM 3-methyl-adenine (3-MA) for 6 h did not restore DUSP4 protein levels (Fig 7G). Furthermore, the inhibitory effect of DOCK1 overexpression on DUSP4 protein was reversed when 20 μM MG132 was added to the cells for 6 h (Fig 7H). These findings strongly suggest that DOCK1 promotes the degradation of the DUSP4 protein primarily via the ubiquitin–proteasome pathway rather than the autolysosome–lysosomal pathway.

### DOCK1 facilitates ubiquitination of DUSP4 protein by regulating HUWE1

Considering the role of DOCK1 in facilitating DUSP4 degradation via the proteasome, we determined its impact on DUSP4 ubiquitination, and observed that the elimination of DOCK1 decreased DUSP4 ubiquitination levels in HTR-8 cells (Fig 8A). Conversely, ectopic expression of DOCK1 in 293T cells enhanced DUSP4 protein ubiquitination (Fig S3E). Considering previous reports implicating the E3 ligase HUWE1 in DUSP4 degradation (Su et al, 2021), we hypothesized that DOCK1 regulates DUSP4 protein degradation by modulating HUWE1. As expected, DOCK1 knockout led to decreased HUWE1 protein expression, and DOCK1 overexpression exerted the opposite effect in HTR-8 cells (Fig 8B). However, when HUWE1 was silenced, DUSP4 protein levels increased with no changes in the transcripts (Fig S3F and G). These findings confirmed that HUWE1 exerts its effects primarily through the regulation of DUSP4 at the protein level. HUWE1 knockdown restored DUSP4 protein stability and decreased DUSP4 ubiquitination levels in cells overexpressing DOCK1 (Fig 8C and D). These results provide compelling evidence that DOCK1 facilitates the degradation of DUSP4 protein through ubiquitination, potentially by modulating HUWE1.

### Treatment with DOCK1 inhibitor TBOPP leads to murine abortion

To study the essential role of DOCK1 in pregnancy, C57BL/6 mice were intraperitoneally injected with TBOPP (0.75 mg per mouse) or vehicle control on GD6, GD8, GD10, and GD12 (Fig 9A). On GD13.5, we assessed the rate of embryo resorption in the TBOPP and vehicle groups. The TBOPP group exhibited a significantly higher rate of abortion than did the vehicle group (Fig 9B and C). To investigate the underlying mechanisms, we analyzed the placental tissues of the mice. After TBOPP treatment, DUSP4 protein levels increased, and the levels of phosphorylated ERK and HUWE1 decreased (Fig 9D

and E). These findings highlight a potentially critical role for DOCK1 in regulating successful pregnancy by modulation of the ERK signaling pathway, with HUWE1 and DUSP4 acting as intermediaries in DOCK1-mediated regulation of successful pregnancy.

In light of the essential role of trophoblast cell invasion and proliferation in ensuring successful murine pregnancy outcomes, we examined the potential influence of TBOPP on these processes by employing CK7 (a marker for trophoblastic cells) immunofluorescence staining in the placenta to observe the region of trophoblast invasion into the decidua. Our results showed that the extent of trophoblast invasion into the decidua was noticeably reduced in the TBOPP treatment group (Fig 9F and G). These data support that the observed embryonic loss under TBOPP treatment might be linked with hindered trophoblast invasiveness. Furthermore, acknowledging the pivotal nature of trophoblast cell proliferation, we next performed immunostaining for Ki67 in placental tissues and found its predominant staining in the labyrinth layer. To analyze the proliferation status, we compared the Ki67 positivity between the TBOPP-treated and control groups. Remarkably, the TBOPP treatment group exhibited a significantly lower percentage of Ki67-positive cells compared with the control group (Fig 9H). These results demonstrate that the DOCK1 inhibitor TBOPP does exert specific effects on trophoblastic cell trophoblastic cell proliferation and invasion, which could then subsequently influence murine pregnancy outcomes.

## Discussion

Placental development is dependent on the regulation of human EVT functions. The trophoblast exhibits characteristics such as high cell proliferation, lack of cell-contact inhibition, migratory and invasive properties, and the ability to evade immune system effectors, especially during the first trimester of pregnancy. The trophoblast is therefore described as a "pseudo-malignant" tissue or a "physiological metastasis" (Ferretti et al, 2007). Abnormal EVT functions are linked to adverse pregnancy outcomes such as RSA and preeclampsia (Fisher, 2015; Grbac et al, 2021). However, the exact cause of the abnormal or shallow placentation associated with these adverse pregnancy outcomes remains uncertain.

The DOCK family of guanine nucleotide exchange factors modulates cytoskeletal dynamics and contributes to cell shape changes, elongation, membrane protrusion, and anterior–posterior polarity. It is also important for EMT and facilitates cell-directed migration (Lamouille et al, 2014; Kukimoto-Niino et al, 2021). After the differentiation of proliferative EVTs into migratory and invasive subtypes, such as intermediate invading EVTs within cell columns, interstitial EVTs, and endovascular EVTs, the behavior of invasive EVTs bears striking similarities to that of transformed cells that exhibit a metastatic phenotype after malignant transformation. DOCK1, a member of the DOCK family, has been implicated in cancer cell metastasis and intestinal epithelial cell migration (Sanders

---

metastasis-related protein levels in HTR-8 cells after DOCK1 knockout or overexpression. **(F)** Western blot results showing EMT or metastasis-related protein levels in JAR cells with silenced or overexpressed DOCK1.

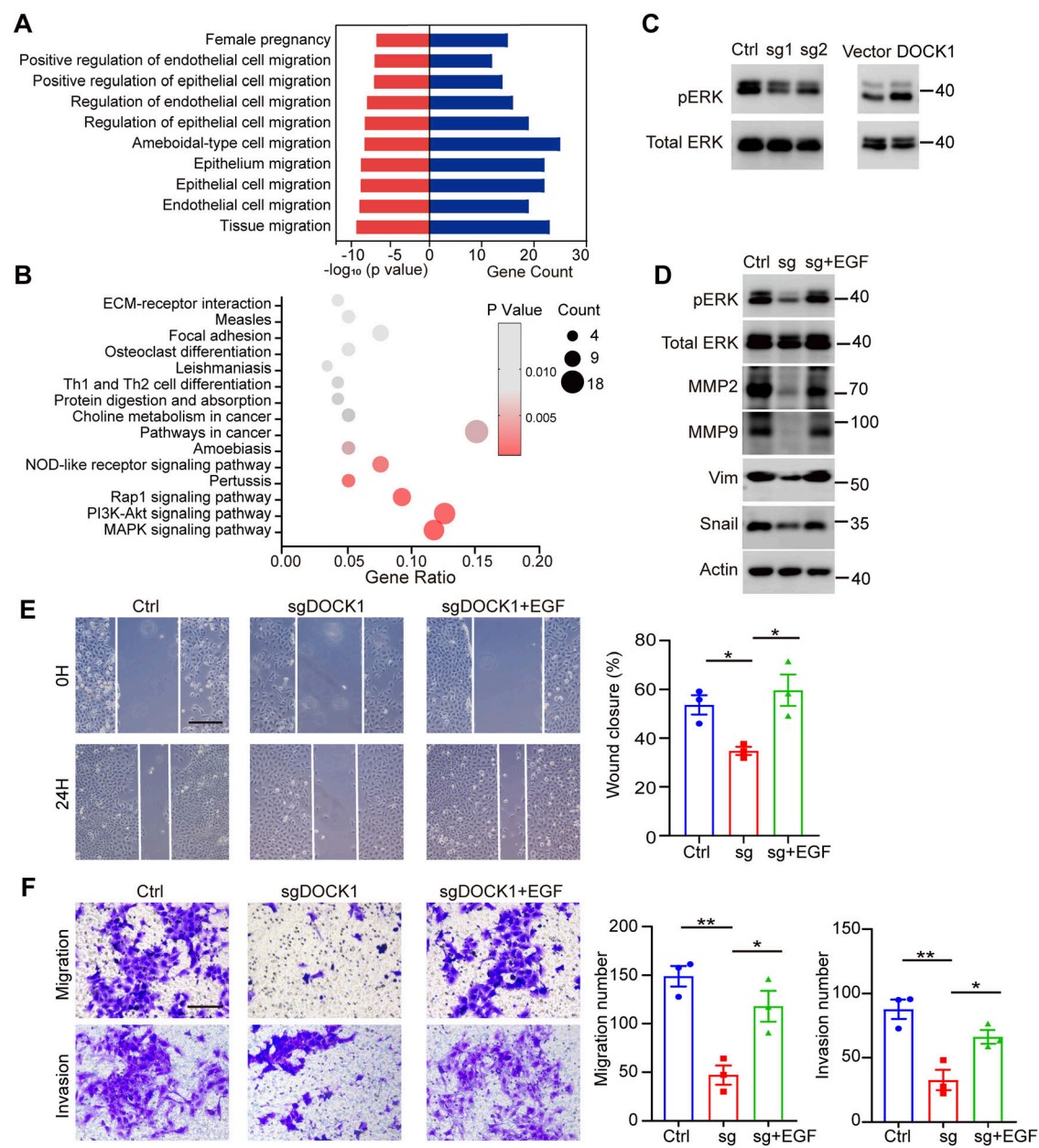

**Figure 5. DOCK1 promotes cell motility via the ERK signaling pathway.**
**(A)** Gene ontology (GO) analysis showing the significant relevance of the top 10 most prominent terms associated with differentially expressed genes (DEGs) after DOCK1 knockout in HTR-8 cells. **(B)** Kyoto Encyclopedia of Genes and Genomes pathway analysis identified the top 15 enriched pathways associated with DEGs. Functional enrichment was used to identify the biological processes associated with the DEGs after DOCK1 knockout in HTR-8 cells. **(C)** Phosphorylation levels of ERK protein in HTR-8 cells after DOCK1 repression or overexpression. **(D)** Metastasis-related protein levels after activation of the ERK signaling pathway in DOCK1 knockout HTR-8 cells. **(E)** Wound healing assay results showing migration ability of HTR-8 cells with DOCK1 knockout after EGF treatment activated the ERK signaling pathway (left). The rate of wound closure was quantified (right). Scale bar: 200 $\mu m$. **(F)** Transwell assays results showing the migration and invasion of HTR-8 cells with DOCK1 knockout after EGF treatment (left). The number of migrating and invading cells was quantified (right). Scale bar: 100 $\mu m$. In (E, F), each group, n = 3 biological replicates (*$P < 0.05$; unpaired two-tailed $t$ test).

et al, 2009; Zhang et al, 2017; Chiang et al, 2019). During human pregnancy, normal EVT function is essential for placental development, and deficient EVT function can lead to pregnancy failure. However, the specific role of DOCK1 in the trophoblast remains largely unexplored. Our data, for the first time, showed that placental villus DOCK1 expression was lower in RSA than in HCs. In

addition, we observed that DOCK1 deficiency impaired trophoblast proliferation and induced cell arrest in the G0/G1 phase.

Trophoblast invasion is essential for the successful establishment of pregnancy, and is a multistep process involving the attachment of trophoblasts to the ECM, degradation of decidual ECM, and migration through eroded connective tissue (Halasz &

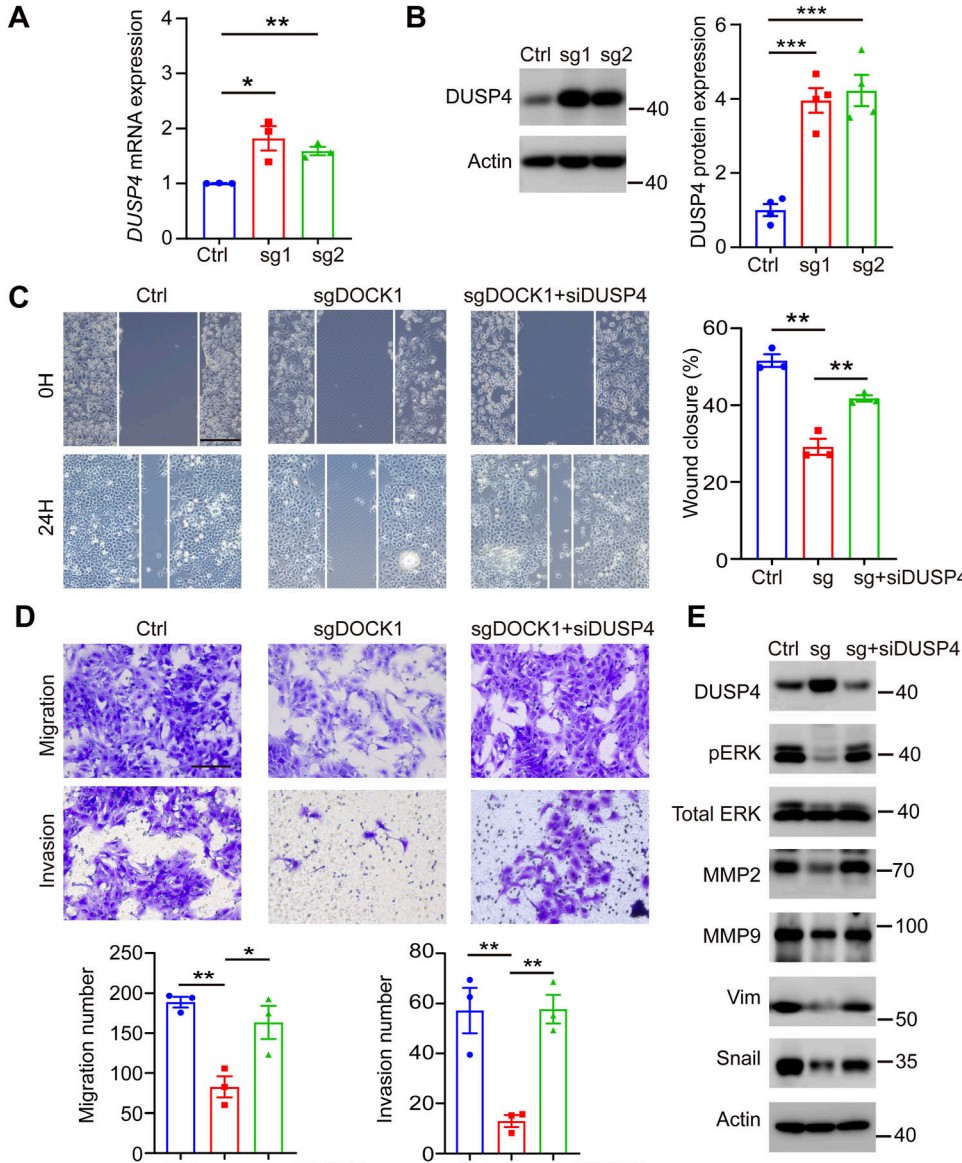

**Figure 6. DOCK1 promotes cell motility via the DUSP4-ERK signaling pathway.**
**(A)** *DUSP4* mRNA expression in DOCK1-knockout HTR-8 cells. **(B)** DUSP4 protein levels in DOCK1-depleted HTR-8 cells (left). The protein levels were quantified (right). **(C)** Wound healing ability after DUSP4 inhibition by siRNA in DOCK1-knockout HTR-8 cells (left). Rate of wound closure was quantified (right). Scale bar: 200 *µm*. **(D)** Migration and invasion ability after DUSP4 inhibition in DOCK1 knockout HTR-8 cells (top). The number of migrating and invading cells was quantified (bottom). Scale bar: 100 *µm*. **(E)** ERK phosphorylation and metastasis-related protein levels after DUSP4 siRNA treatment in DOCK1-knockout HTR-8 cells. In (A, C, D), each group, n = 3 biological replicates (*$P < 0.05$; **$P < 0.01$; unpaired two-tailed $t$ test). In (B), each group, n = 4 biological replicates (***$P < 0.001$; unpaired two-tailed $t$ test).

Szekeres-Bartho, 2013). EVTs express metalloproteinases (MMPs), which are composed of at least 17 zinc-dependent endopeptidases that degrade the endometrial ECM during trophoblast invasion (Estella et al, 2012). MMP2 and MMP9 are among the best characterized MMPs. In the placental tissue during early pregnancy, MMP2 is mostly expressed in EVT cells, and MMP9 is expressed in villous CTBs and EVT cells (Isaka et al, 2003). In our study, we found that DOCK1 depletion reduces the levels of MMP2 and MMP9.

After blastocyst adherence to the endometrium, trophoblasts undergo the EMT process. Trophoblasts are initially characterized as epithelial cells and subsequently switch to mesenchymal, allowing their migration into the maternal decidua. During this process, the expression of EMT-related molecules changes, and associated signaling pathways are activated (Imakawa et al, 2018). For instance, upregulation of the transcriptional factor snail contributes to the

activation of the mesenchymal phenotype and is involved in the initiation and progression of EMT (Lamouille et al, 2014). Cells acquiring anterior–posterior polarity and reorganizing their cytoskeletal structure are important EMT processes that enhance cell motility and invasion (Lamouille et al, 2014). Many cytoskeletal proteins, including DOCK1, facilitate this transition. In our study, DOCK1 promoted cell migration and invasion. Moreover, DOCK1 regulated the expression of metastasis-related marker proteins, and the morphology of trophoblasts changed from spindle to round after DOCK1 depletion, proving that the process of EMT is disrupted.

In addition, DOCK1 is critical for inducing β-catenin translocation into the nucleus, which enhances trophoblast motility. β-catenin translocating from the cytoplasm into the nucleus can interact with the transcription factors lymphoid enhancer factor or T-cell factor to initiate the transcription of a broad spectrum of downstream

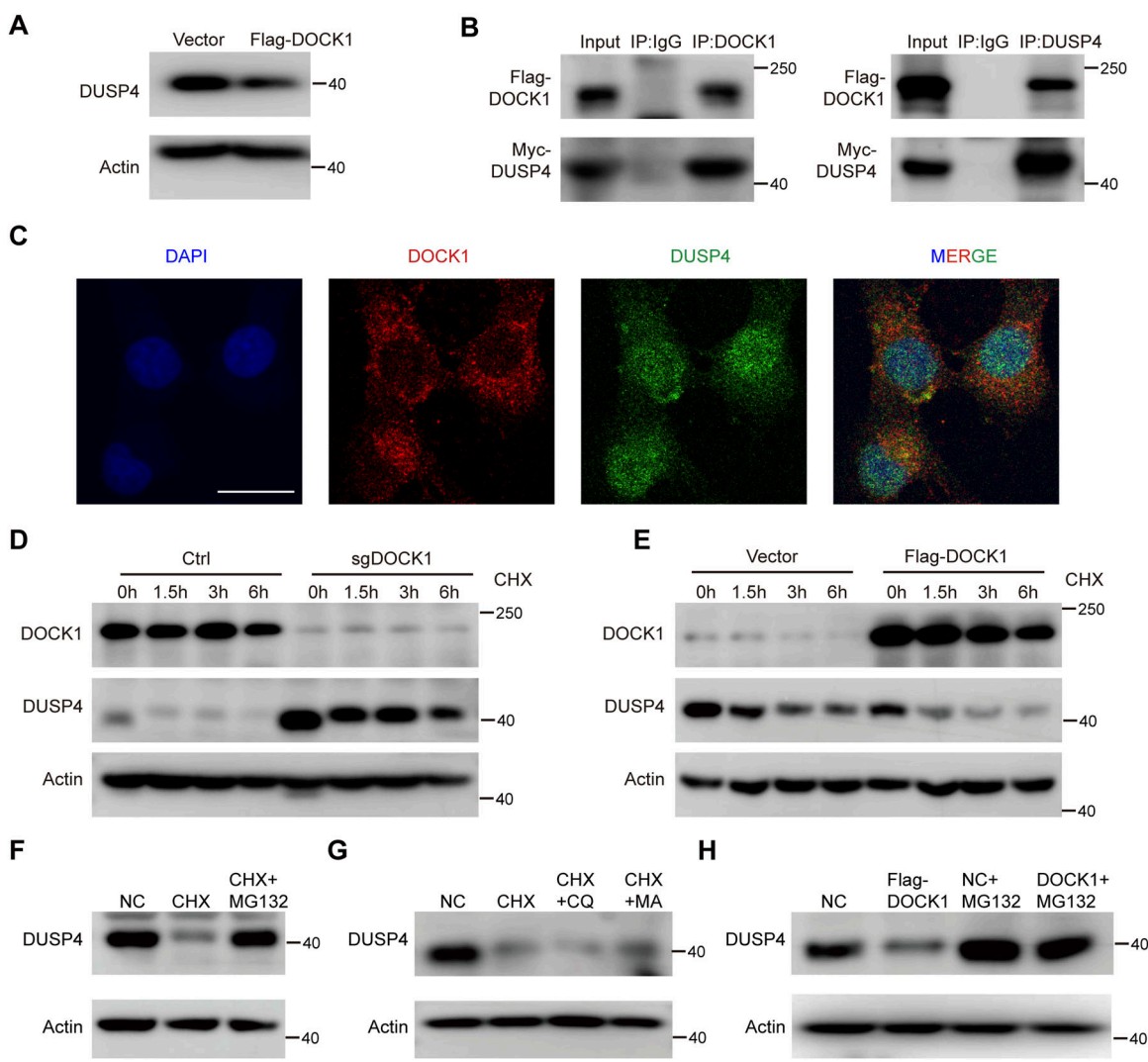

**Figure 7. DOCK1 interacts with DUSP4 and impairs its stability.**
**(A)** DUSP4 protein levels in DOCK1-overexpressing HTR-8 cells. **(B)** Interaction of DOCK1 and DUSP4 detected by CO-IP in HEK293T cells transfected with ectopic Flag-DOCK1 and Myc-DUSP4. **(C)** Interaction of DOCK1 and DUSP4 in the cytoplasm observed by immunofluorescence staining with anti-DOCK1 (red) and anti-DUSP4 (green) in HTR-8 cells. Scale bar: 25 μm. **(D)** DOCK1 knockout HTR-8 cells were treated with 50 μg/ml cycloheximide (CHX) for 0, 1.5, 3, and 6 h, and DOCK1 and DUSP4 proteins were assayed by Western blot. **(E)** HTR-8 cells overexpressing DOCK1 were treated with 50 μg/ml CHX for 0, 1.5, 3, and 6 h, and Western blot was used to detect DOCK1 and DUSP4 proteins. **(F)** HTR-8 cells were treated with 50 μg/ml CHX alone or 50 μg/ml CHX and 20 μM MG132 for 6 h. **(G)** HTR-8 cells were treated with 50 μg/ml CHX alone or 50 μg/ml CHX and 10 μM CQ or 50 μg/ml CHX plus 1 mM 3-MA for 6 h. **(H)** DOCK1-overexpression and control HTR-8 cells were treated with 20 μM MG132 or left untreated for 6 h.

genes involved in cell metastasis (Yuan et al, 2021). The differentiation of CTB into aggressive EVT is associated with the nuclear recruitment of β-catenin. In addition, β-catenin can activate a series of molecules that promote invasion and EMT-like features in migrating trophoblasts (Knöfler & Pollheimer, 2013).

The MAPK-ERK signaling pathway is crucial for placenta formation and the activated ERK pathway is essential for trophoblast migration, invasion, cytoskeletal dynamics, and integrin switching (Gupta et al, 2016). Our RNA-seq results showed the involvement of DOCK1 in the MAPK signaling pathway and regulation of ERK phosphorylation. Furthermore, we observed significant up-regulation of DUSP4 protein, a negative regulator of the MAPK signaling pathway, and subsequent repression of trophoblast motility after DOCK1 depletion. In addition, DOCK1 modulated the stability of DUSP4 protein, suggesting

that DOCK1 regulates DUSP4 expression by posttranslational protein modifications. DUSP4 is degraded by a ubiquitin-mediated proteasomal pathway, and its enhanced stability inhibited ERK activity (Gómez et al, 2013). Consistently, our study unveiled the novel binding between DOCK1 and DUSP4, and the depletion of DOCK1 suppressed DUSP4 ubiquitination and protein degradation. These findings highlight the pivotal role of this interaction in regulating trophoblast functions.

The mechanism by which DOCK1 modulates the ubiquitination of DUSP4 through ubiquitin-conjugating or deubiquitinating enzymes merits further investigation. The E3 ligase enzyme HUWE1 contributes to the degradation of DUSP4 through ubiquitination (Su et al, 2021). Interestingly, we found that HUWE1 acts as a key E3 ubiquitin ligase partner of DOCK1 where in it mediates the

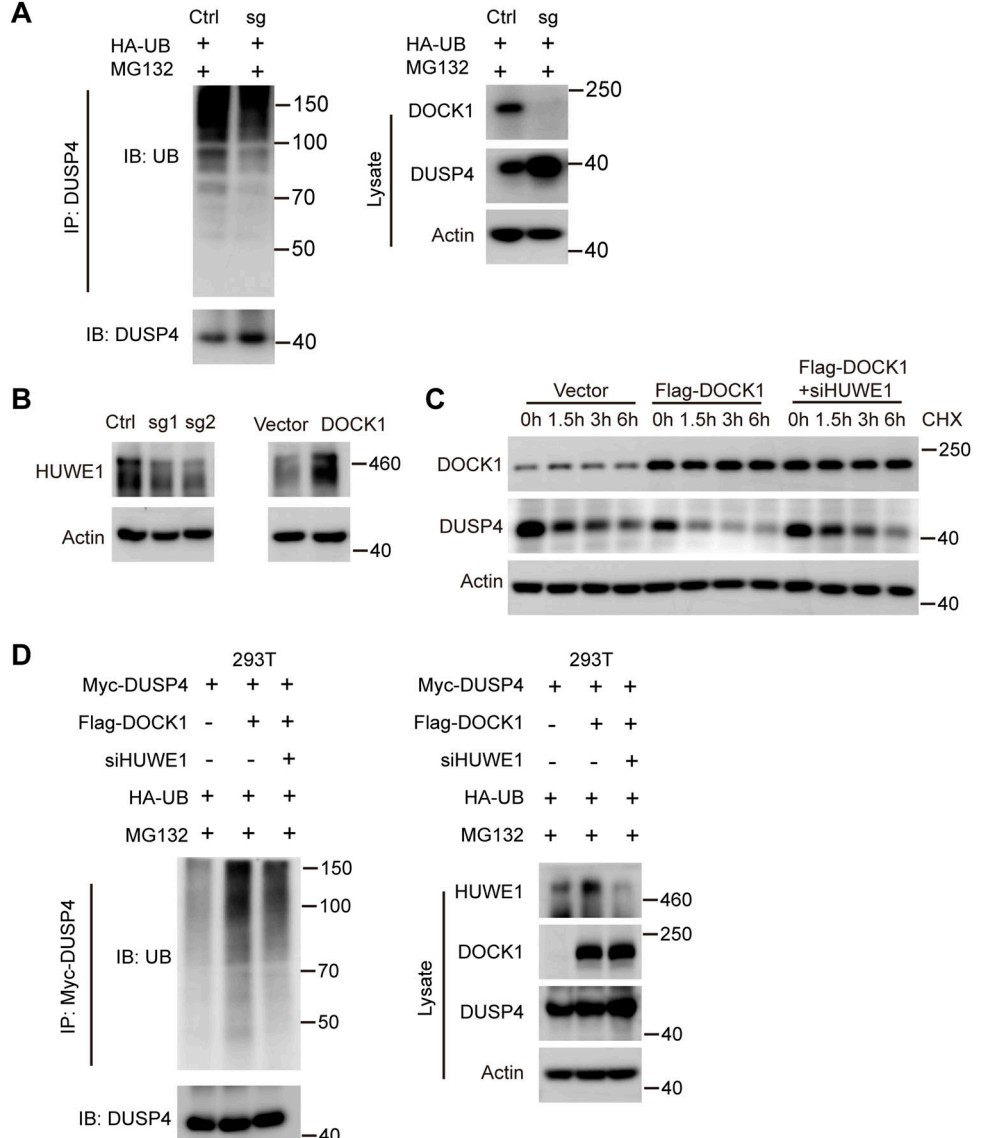

**Figure 8. DOCK1 promoted DUSP4 degradation via the ubiquitin proteasomal pathway.**
**(A)** DOCK1-knockout HTR-8 cells were transfected with HA-UB for 48 h and treated with 20 $\mu$M MG132 for 6 h. The cell lysates were immunoprecipitated with the DUSP4 antibody, and the ubiquitination assay showed the ubiquitination levels of DUSP4 protein. **(B)** HUWE1 protein levels after DOCK1 knockout or overexpression in HTR-8 cells. **(C)** After DOCK1 overexpression followed by HUWE1 knockdown, HTR-8 cells were treated with 50 $\mu$g/ml cycloheximide for 0, 1.5, 3, and 6 h, and Western blot was performed to detect DOCK1 and DUSP4 protein levels. **(D)** 293T cells were transfected with ectopic Flag-DOCK1, Myc-DUSP4, and HA-UB for 24 h, and then transfected with HUWE1 siRNA for 48 h. Finally, 20 $\mu$M MG132 was added and the protein was extracted after 6 h of incubation. The cell lysates were immunoprecipitated with Myc antibody and the ubiquitination assay showed DUSP4 protein ubiquitination levels.

ubiquitination and degradation of DUSP4 protein. However, it is unclear whether and which lysine residues (Lys6, Lys11, Lys27, Lys29, Lys33, Lys48 or Lys63) or the $\alpha$-amino group of the N-terminal methionine (Met1) in ubiquitin promote the formation of poly-ubiquitin chains on DUSP4 protein. In addition, the specific binding interface and the essential regions of these genes that affect their interaction need to be determined. Further investigations are required to elucidate the mechanisms underlying the interaction between DOCK1 and HUWE1 and their impact on DUSP4, to provide insights into the regulation of cell motility.

Our results showed that DOCK1 regulates their proliferation, migration, and invasion capabilities of trophoblasts. Suppressing DOCK1 led to a decline in trophoblast cell proliferation and a reduction in the extent of trophoblast invasion into the decidua in vivo. These findings demonstrated that the impact of DOCK1 inhibition on trophoblastic cell functionalities, with potential ad-

verse effects on murine pregnancy outcomes. Yet, a pertinent gap in our understanding is the direct influence of DOCK1 on embryonic cells; we will delve deeper into understanding the role of both DOCK1 and TBOPP in the broader spectrum of fetal development in our future study. This integrated approach ensures a well-rounded understanding, potentially addressing critical questions and offering therapeutic insights into pregnancy complications like RSA.

# Materials and Methods

### Patient sample collection

The first-trimester villous tissues were collected from patients with unexplained RSA group and patients who terminated their

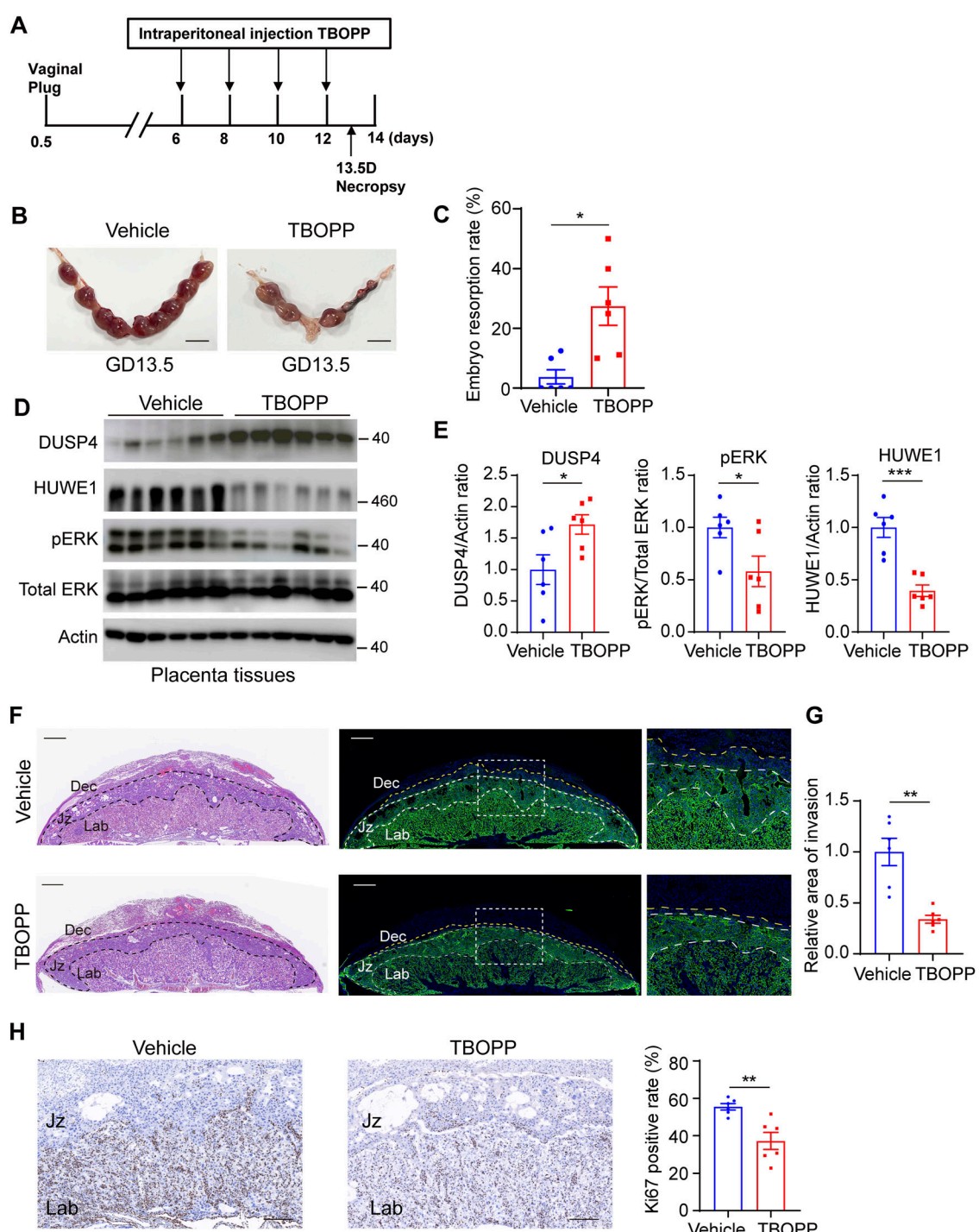

**Figure 9. DOCK1 deficiency triggers embryo miscarriage in pregnant mice models.**
**(A)** Schematic illustration of the protocol for inhibiting DOCK1 during murine pregnancy. Experimental mice received vehicle control or TBOPP on GDs 6, 8, 10, and 12 through intraperitoneal injections, and were euthanized on GDs 13.5. **(B)** Morphology of pregnant uteri with embryo implantation sites on GD 13.5 in mouse models after vehicle control or TBOPP treatments (n = 6 mice for each group). Scale bar: 1 cm. **(C)** Embryo resorption rate was calculated. **(D)** Western blot assays were performed to detect the levels of DUSP4, phosphorylated ERK, and HUWE1 after TBOPP treatment. **(E)** Quantification of DUSP4, phosphorylated ERK, and HUWE1 protein levels. **(F)** Representative HE-stained images of placentas and CK7 immunofluorescence of trophoblast cell invasion into the decidua after TBOPP treatment (left). Based on the HE and CK7 staining, distinct layers—Dec (decidua), Jz (junctional zone), and Lab (labyrinth)—are delineated by black lines in HE images and white lines in immunofluorescence images. Boxes on the left correspond to magnified sections on the right, with the region between the yellow and white lines indicating the trophoblast cell invasion area into the decidua. Scale bar: 500 $\mu$m. **(G)** The area of trophoblast cell invasion into the decidua was quantified. **(H)** Representative images of Ki67 immunostaining in placental tissues after TBOPP treatment (left). Percentage of Ki67-positive cells was calculated (right). Scale bar: 200 $\mu$m. In (C, E, G, H), each group, n = 6 mice (*$P < 0.05$; **$P < 0.01$; ***$P < 0.001$, unpaired two-tailed $t$ test).

pregnancies for nonmedical reasons (HC group). To ensure comparability between the groups, we matched the gestational weeks and ages of the pregnant women in both groups. Informed consent was obtained from all patients. RSA was defined as having experienced two or more successive abortions, and individuals with the following health problems were excluded from the study: (1) parental or fetal chromosome abnormalities, (2) endocrine or metabolic diseases (polycystic ovarian syndrome, diabetes mellitus, hyperprolactinemia, hypothyroidism, hyperthyroidism or luteal phase deficiency), (3) uterine or cervical anatomical abnormalities, (4) infectious diseases (including AIDS, tuberculosis, gonorrhea, and syphilis), (5) autoimmune disease (such as systemic lupus erythematosus, antiphospholipid antibody syndrome, Sjogren's syndrome, and systemic sclerosis). The collected placental tissues were immediately separated into villi and decidua, and the villus samples were washed with PBS and cut into small pieces of approximately of $1 \times 1 \times 1$ $cm^3$. For further analysis, the villous tissues were either immediately frozen in liquid nitrogen and stored at −80°C for Western blot and mRNA extraction or fixed in 4% PFA and embedded in paraffin for immunofluorescence staining. All experiments were approved by the Medical Ethics Committee of the International Peace Maternity and Child Health Hospital of China Welfare Institute (No.: GKLW2021-17) (Shanghai, China).

## Cell culture

The human trophoblast cell line HTR-8/SVneo, derived from chorionic villi explants of the first-trimester placenta were a gift from Dr. PK Lala (University of Western Ontario, Canada), and were stored in our laboratory. The choriocarcinoma-based trophoblast cell line JAR and HEK293T cell lines were purchased from Shanghai Cell Bank, Chinese Academy of Sciences (Shanghai, China). HTR-8/SVneo was cultured in DMEM/F12 (Gibco) with 10% FBS (Gibco) and 1% penicillin/streptomycin (Invitrogen). JAR and 293T cells were cultured in high-glucose DMEM with 10% FBS with 1% penicillin/streptomycin. Both cell lines were maintained in a humidified atmosphere at 37°C with 5% $CO_2$. Cells were routinely checked to confirm the absence of mycoplasma, and cell stocks were authenticated.

## Western blot analysis

To obtain total protein lysates, cells and tissues were lysed using RIPA buffer (Thermo Fisher Scientific). For the separation of nuclear and cytoplasmic protein, the Nuclear and Cytoplasmic Extraction Kit (Thermo Fisher Scientific) was used according to the manufacturer's instructions. Protease inhibitors, phosphatase inhibitors (Thermo Fisher Scientific), and 5× loading buffer were then added to the lysates. Proteins were separated by electrophoresis using SDS–PAGE and transferred onto 0.22 or 0.45-$\mu$m PVDF membranes, depending on the molecular weight of the proteins. After blocking with 5% defatted milk at RT for 1 h, the membranes were incubated overnight at 4°C with primary antibodies. The membranes were then washed three times with TBST and incubated with the appropriate secondary antibody for 1 h at RT. Protein bands were detected using enhanced chemiluminescence reagent (Millipore) and visualized using the Amersham Imager 600 (GE Healthcare Life

Science). Relative protein expression levels were quantified using Image J software. Details of the primary and second antibodies used in this study are provided in Table 1.

## RNA extraction and qRT–PCR

The total RNA from tissues and cells as extracted using TRIzol reagent (Thermo Fisher Scientific), and reverse-transcribed into cDNA by using the PrimeScript RT reagent kit (Takara). qRT–PCR was performed using the SYBR Green Kit (Takara) in the Real-Time PCR System (Applied Biosystems) to determine the mRNA expression levels of target genes. The $2^{-\Delta\Delta CT}$ method was applied to calculate the relative mRNA expression. The primer sequences used are listed in Table 2.

## Plasmid and siRNA

To generate cells overexpressing DOCK1 or DUSP4, trophoblasts were transfected using jetPRIMEPolyplus-transfection by exogenous plasmid introduction. DOCK1 exogenous plasmid was purchased from Youbio and HUWE1 exogenous plasmid was obtained from Addgene. DOCK1 or HUWE1 or DUSP4 transient knockdown cells were transfected with siRNA of the required genes from GenePharma using Oligofectamine reagent (Thermo Fisher Scientific). The sequences used are presented in Table 3. To construct stable DOCK1 knockout cell lines, pSpCas9(BB)-2A-Puro-sgDOCK1 and its control vector were transfected into HTR-8 cells by using jetPRIMEPolyplus-transfection. Transfected cells were selected with puromycin (2 $\mu$g/ml) for 1 wk, and the surviving cells were picked and seeded into 96-well plates to form cell clones and expand until they could be verified by Western blot. The DOCK1-sgRNA sequences used are shown in Table 3.

## RNA-seq analysis

Total RNA was extracted from HTR-8/SVneo cell knockout DOCK1 and its controls using 1 ml of TRIzol (Thermo Fisher Scientific) and transferred to Novogene. The sequencing library was constructed after rRNA removal using the NEB Next Ultra RNA Library Prep Kit for Illumina (NEB). After the sequencing library was qualified, reads were mapped to the reference genome using the DEseq2 R package for normalization and identification of DEGs. DEGs were applied for gene ontology (GO) and Kyoto Encyclopedia of Genes and Genomes analysis using a threshold value of $P < 0.05$ and $|log_2$ foldchange$| > 1$.

## Cell proliferation

Trophoblast cells (HTR-8 and JAR) were seeded in 96-well plates at a density of $2 \times 10^4$ cells/ml and $3 \times 10^4$ cells/ml, respectively. After 0, 24, 48, 72 or 96 h, cell counting kit-8 (CCK-8) solution (Yeason) was added into to each well for 1.5 h at 37°C, and the absorbance values were measured at the wavelength of 450 nm using a microplate reader. For the colony formation assay, HTR-8 (1,000 cells/well) and JAR (1,500 cells/well) cells were inoculated into a six-well plate, respectively, and cultured cells for 10 d. The colonies were stained with 0.5% crystal violet (Yeason), and the number of colonies was counted. For the EdU (5-ethynyl-2′-deoxyuridine) assay, the

**Table 1. Details of antibodies used.**

| Antibody | Application (dilution) | Source | Catalog number |
|---|---|---|---|
| DOCK1 | WB(1:1,000) | CST | #4846 |
| DOCK1 | IHC(1:800), IF(1:200) | Abcam | #ab97325 |
| DUSP4 | WB(1:1,000) | Abcam | #ab222487 |
| DUSP4 | IF(1:200) | Proteintech | #66349-1-Ig |
| Snail | WB(1:1,000) | CST | #3879 |
| Vimentin | WB(1:1,000) | CST | #5741 |
| MMP9 | WB(1:1,000) | CST | #13667 |
| MMP2 | WB(1:1,000) | CST | #87809 |
| N-cadherin | WB(1:1,000) | CST | #13116 |
| E-cadheirn | WB(1:1,000) | CST | #14472 |
| HUWE1 | WB(1:1,000) | Proteintech | #19430-1-AP |
| Lamin B1 | WB(1:1,000) | Proteintech | #66095-1-lg |
| β-Actin | WB(1:2,000) | CST | #4970 |
| Phospho-p44/42 MAPK (Erk1/2) (Thr202/Tyr204) | WB(1:1,000) | CST | #4370 |
| p44/42 MAPK (Erk1/2) | WB(1:1,000) | CST | #4695 |
| β-Catenin | WB(1:1,000) | CST | #8480 |
| Ubiqutin | WB(1:1,000) | CST | #3936 |
| Ki67 | IHC(1:500) | Abcam | #ab15580 |
| CK7 | IF(1:200) | Abcam | #ab181598 |
| HLA-G | IF(1:200) | Proteintech | #66447-1-Ig |
| Anti-rabbit IgG (conformation specific) | WB(1:2,000) | CST | #5127 |
| Goat anti-mouse IgG (H+L), HRP conjugate | WB(1:5,000) | Proteintech | #SA00001-1 |
| Goat anti-rabbit IgG (H+L), HRP conjugate | WB(1:5,000) | Proteintech | #SA00001-2 |

**Table 2. Primers used for real-time quantitative polymerase chain reaction.**

| Gene | Forward primer (5′-3′) | Reverse primer (5′-3′) |
|---|---|---|
| DOCK1 | ACCGAGGTTACACGTTACGAA | TCGGAGTGTCGTGGTGACTT |
| DOCK2 | AGCACAAAATGTTACAGGGCA | CCATCAGATCGTACATCATGGAC |
| DOCK3 | TATGCAGCTTTCGAGGATCTGT | GCCCATTCTTGTAGAGTTGCT |
| DOCK4 | ATGTGGATACCTACGGAGCAC | CCAATTTCCAATGACAGGCCATA |
| DOCK5 | GAGGCAGAAGTACGGGGTTG | CAGGAATCACGGTTTCATGCT |
| DOCK6 | GCTTCTGGAGACGAGAGGTC | AGGTTCCTCAGGTCGAAGATG |
| DOCK7 | GTGGCAGCCGAAGTTAGGAAG | GCACTGTGGTGTGATGGGATA |
| DOCK8 | TTCACGCCAAAGGAATGTAGG | GTCCCTGACATGAGGGTCCA |
| DOCK9 | TGTCATCGTCCAGAAGAAGACT | TCTCAGGATGGCCGTCTGAAA |
| DOCK10 | TCTGGAGACATTCGACAGCTA | TGGGAAGTGGTATCTTCATCCT |
| DOCK11 | CTTGGGCCAAATTGGAGACAA | CCCACAGCATTTCACGGAC |
| DUSP4 | GGCGGCTATGAGAGGTTTTCC | TGGTCGTGTAGTGGGGTCC |
| HUWE1 | TTGGACCGCTTCGATGGAATA | TGAAGTTCAACACAGCCAAGAG |
| β-Actin | CATGTACGTTGCTATCCAGGC | CTCCTTAATGTCACGCACGAT |

trophoblasts were seeded in a 96-well plate. When the cells were in the logarithmic growth phase, the Edu working solution was introduced into the cells, and they were cultured for 1.5 h at 37°C. The cells were fixed and stained according to the manufacturer's instructions (C10310-1; RIBOBIO). Images were taken under a microscope, and the number of positive cells was counted.

## Cell cycle

HTR-8 and JAR cells, seeded at a density of $1 \times 10^6$ cells in 6-cm dishes, were collected and permeabilized with prechilled 70% ethanol at −20°C overnight. Subsequently, cells underwent RNase A treatment and followed by propidium iodide staining in the dark at RT for 30 min (CCS012; MULTI SCIENCES). Flow cytometric analysis was performed immediately and the DNA histograms were evaluated using the FlowJo software to determine cell cycle distribution.

## Scratch wound-healing assay

For the scratch wound-healing assay, the trophoblasts were placed into a six-well plate. When the cell confluence reached 80–90%,

**Table 3. Sense sequences selected for knockdown and knockout.**

| Gene | Sense sequences (5'-3') | Antisense sequences (5'-3') |
|---|---|---|
| siDOCK1-1 | GGCCCAAUAUUAGAAAUGATT | UCAUUUCUAAUAUUGGGCCTT |
| siDOCK1-2 | GGUCCAGCGAUUUGAAUAUTT | AUAUUCAAAUCGCUGGACCTT |
| siDUSP4-1 | CUAUCAGUACAAGUGCAUCTT | GAUGCACUUGUACUGAUAGTT |
| siDUSP4-2 | UCUUCAGCUUUCCGGUCUCTT | GAGACCGGAAAGCUGAAGATT |
| siHUWE1-1 | GAGUUUGGAGUUUGUGAAGUUTT | AACUUCACAAACUCCAAACUCTT |
| siHUWE1-2 | GCUCCCACUAUAACCUCACTT | GUGAGGUUAUAGUGGGAGCTT |
| sgDOCK180-1 | CACCGAATAGCTTCTAAACAAGTGG | AAACCCACTTGTTTAGAAGCTATTC |
| sgDOCK180-2 | CACCGCTAATCGTGCTAGTTAATTC | AAACGAATTAACTAGCACGATTAGC |

a physical gap was made using a pipette tip. The cells were washed with PBS and cultured in 1% serum medium. At 0 and 24 h, the area of the scratch wound was photographed and quantified using Image J software.

### Transwell assay

$5 \times 10^4$ HTR-8 and $3 \times 10^4$ cells JAR cells were placed in transwell plates (8 µm pore size) in a 24-well plate with a serum-free medium. The lower chambers were filled with medium containing 20% FBS. After incubation for 24 h at 37°C, the migrated cells on the lower surface of the inserts were fixed with 4% PFA and stained with 0.5% crystal violet. For invasion assay, $1.5 \times 10^5$ HTR-8 and $1 \times 10^5$ JAR cells were seeded in membrane inserts with Matrigel (BD Biosciences). After 24 h, the invading cells were fixed and stained. Finally, images were acquired under a microscope, and the number of cells was counted.

### Animal experiments

For in vivo experiments, 7–8-wk-old female and male C57BL/6 mice were purchased from Shanghai Laboratory Animal Center, Chinese Academy of Science. The mice were allowed to acclimate for 1 wk in a specific pathogen-free room. Subsequently, the female and male mice were paired overnight. On appearance of a vaginal plug, the following day was designated as day 0.5 of gestation. The pregnant mice were randomly divided into two groups. The vehicle control group was intraperitoneally administrated with 150 µl saline containing 10% DMSO (D8418; Sigma-Aldrich) and 20% Cremophor EL (HY-Y1890; MedChemExpress) on days 6, 8, 10, and 12 of gestation. The TBOPP group was intraperitoneally administrated with 150 µl saline containing the DOCK1 inhibitor TBOPP (HY-124711; MedChemExpress) dissolved in saline containing 10% DMSO and 20% Cremophor EL on days 6, 8, 10, and 12 of gestation at 0.75 mg per mouse. Female mice were euthanized on day 13.5, and the embryo absorption rate was counted. The embryo absorption rate was calculated as the ratio of the number of resorption fetuses (hemorrhagic implantations)/(viable surviving fetuses + resorption fetuses). The animal experiments were approved by the Medical Ethics Committee of the International Peace Maternity and Child Health Hospital of China Welfare Institute (No.: GKLW2021-17) (Shanghai, China).

### Hematoxylin and eosin (HE), immunohistochemistry, and immunofluorescence staining

Freshly collected tissues were fixed in 4% PFA, embedded in paraffin and then sectioned at 4 µm thickness. HE staining was performed by staining the sections with hematoxylin followed by eosin. For immunofluorescence staining, antigen retrieval, and serum blocking were performed before incubating with primary antibodies at 4°C overnight. The sections were subsequently incubated with secondary antibodies conjugated to a fluorophore and counter-stained with DAPI (Abcam). Slides were observed under a fluorescence microscope. For immunohistochemical analysis, sections were incubated with primary antibodies overnight at 4°C, then with corresponding biotinylated secondary antibodies. Subsequent to secondary antibody incubation, the sections were treated with 3,3'-DAB substrate for chromogenic detection. Positive staining was visualized as brown precipitate under a light microscope. Quantitative analyses were performed using Image J software: The Ki67 positivity rate was determined by calculating the ratio of Ki67-positive cells to total cells, and the extent of trophoblast into the decidua, as indicated by CK7 staining, was also assessed. For cell immunofluorescence assays, trophoblasts were cultured in a six-well plate containing sterile coverslips. Upon reaching the logarithmic phase, cells were fixed with 4% PFA for 10 min and washed with PBS. The slides were treated with 0.3% TritonX-100 for 10 min and washed with PBS. After incubating with primary antibodies overnight at 4°C, the slides were washed with PBS and incubated with fluorescent secondary antibodies for 1 h. Details of the primary and second antibodies used are provided in Table 1.

### Statistical analysis

All statistical analyses and figures displayed were generated using GraphPad Prism 8.0 (GraphPad Software). The unpaired two-tailed $t$ test was used to perform statistical comparisons between two groups. Data were presented as mean ± SEM. A $P$-value < 0.05 was considered statistically significant.

## Data Availability

The RNA-seq data from this publication were deposited to the GEO database and assigned the identifier GSE232808.

# Supplementary Information

# Acknowledgements

We extend our sincere gratitude to Professor Xueqiong Zhu and her team at the Second Affiliated Hospital of Wenzhou Medical University for their invaluable insights. We acknowledge the support of the National Key Research and Development Program of China (grant number 2018YFC1002800), the National Natural Science Foundation of China (grant number 82171669 and 81971403), and the Funds for Outstanding Newcomers, Shanghai Sixth People's Hospital (grant number X-3664) to Y Lin; Innovative Research Team of High-level Local Universities in Shanghai (grant number SHSMU-ZLCX20210202) and Shanghai Jiao Tong University Trans-Med Awards Research (STAR) (Major Project) (grant number 20210201) to Y Lin and C Chen; Natural Science Foundation of Shanghai (22ZR1467700), Shanghai Pujiang Program (22PJD083) to W Zeng.

## Author Contributions

Y Xu: conceptualization, data curation, formal analysis, validation, investigation, visualization, methodology, and writing—original draft, review, and editing.
X Liu: resources, investigation, methodology, project administration, and writing—review and editing.
W Zeng: investigation, methodology, and writing—review and editing.
Y Zhu: validation, investigation, visualization, and methodology.
J Dong: investigation, visualization, and methodology.
F Wu: data curation, formal analysis, and writing—review and editing.
C Chen: formal analysis, visualization, and methodology.
S Sharma: conceptualization, investigation, and writing—review and editing.
Y Lin: conceptualization, resources, data curation, supervision, funding acquisition, methodology, project administration, and writing—review and editing.

## Conflict of Interest Statement

The authors declare that they have no conflict of interest.

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
