## [Reviewer comments · Life Science Alliance]

Life Science Alliance

DOCK1 insufficiency disrupts trophoblast function and pregnancy disorders via DUSP4-ERK pathway

Yichi Xu, Xiaorui Liu, Weihong Zeng, Yueyue Zhu, Junpeng Dong, Fan Wu, Cailian Chen, Surendra Sharma, and Yi Lin
DOI: <https://doi.org/10.26508/lsa.202302247>

Corresponding author(s): Yi Lin, Shanghai Jiao Tong University School of Medicine Affiliated Sixth People's Hospital

Review Timeline:

Submission Date:	2023-06-30
Editorial Decision:	2023-09-11
Revision Received:	2023-10-29
Editorial Decision:	2023-11-03
Revision Received:	2023-11-06
Accepted:	2023-11-07

Transaction Report:

September 11, 2023

Re: Life Science Alliance manuscript #LSA-2023-02247-T

Yi Lin
Shanghai Jiao Tong University
Shanghai, China

Dear Dr. Lin,

Thank you for submitting your manuscript entitled "DOCK1 insufficiency disrupts trophoblast function and pregnancy disorders via DUSP4-ERK pathway" to Life Science Alliance. The manuscript was assessed by expert reviewers, whose comments are appended to this letter. We invite you to submit a revised manuscript addressing the Reviewer comments.

Thank you for this interesting contribution to Life Science Alliance. We are looking forward to receiving your revised manuscript.

Sincerely,

B. MANUSCRIPT ORGANIZATION AND FORMATTING:

Reviewer #1 (Comments to the Authors (Required)):

In the present study, authors investigated the role of DOCK1 in trophoblast function and pregnancy outcomes, uncovering its association with recurrent spontaneous abortion (RSA) and extravillous trophoblasts (EVTs). The deficiency of DOCK1 disrupts DUSP4 degradation, thereby affecting the ERK pathway, EVT migration, and invasion. Inhibiting DOCK1 leads to miscarriage through the DUSP4/ERK pathway, underscoring its significance in addressing pregnancy disorders like RSA and preeclampsia. The manuscript is effectively written, engaging, and easy to comprehend. Overall, the experimental design, methodology, statistical analysis, and presentation of results provide compelling support. Nevertheless, there are concerns regarding certain aspects of the manuscript, which are elaborated below: Was there any impact on the cell cycle/polyploidization due to the inhibition of DOCK1 or TBOPP? Figure 3: Kindly verify the expression of DOCK1 in cytosolic and nuclear fractions of HTR8 and JAR cells.

Reviewer #2 (Comments to the Authors (Required)):

This is a well-written manuscript and appropriately designed study looking at the role of DOCK1 pathway associated with RSA. Authors found DOCK1 is reduced at the trophoblast compartment of the RSA placenta, therefore, used HTR8 cell line to characterize cellular functions. Authors derived HTR8 lines with overexpressed and knockdown of DOCK1, and characterized cell proliferation and invasion and found DOCK1 overexpression increases degradation of the DUSP through ubiquitin-proteasome system, leading to inactivation of ERK signaling pathways. Authors also performed in vivo experiments using DOCK1 inhibitor TBOPP, and found that mice treated with TBOPP showed increased rate of embryo resorption compared to the control. Overall, the manuscript is well-written. I have very little comments for this manuscript summarized below.

- 1) Figure 6C is difficult to see. DAPI is faint and cannot tell which are nuclear and which are cytoplasmic staining.
- 2) In vivo mouse experiments is very exciting, however, authors only examined the protein expression of key markers, DUSP4, HUWE1, ERK etc, that authors have studied across the manuscript. The issues I have is that whether the treatment of DOCK1 inhibitor is mimicking the RSA pathophysiology or just killing the cells and led to embryo resorption. Just from this experiment, authors cannot rule out/draw the conclusion whether the drugs have the effect on the embryo development, or the actual trophoblastic abnormalities led to the abortion. Authors should consider looking into other cellular phenotypes such as cell proliferation and cell invasion in the mouse placenta to confirm the cellular phenotype is also seen in the TBOPP cases.

Reviewers' Comments to Author:

Reviewer #1 (Comments to the Authors (Required)):

In the present study, authors investigated the role of DOCK1 in trophoblast function and pregnancy outcomes, uncovering its association with recurrent spontaneous abortion (RSA) and extravillous trophoblasts (EVTs). The deficiency of DOCK1 disrupts DUSP4 degradation, thereby affecting the ERK pathway, EVT migration, and invasion. Inhibiting DOCK1 leads to miscarriage through the DUSP4/ERK pathway, underscoring its significance in addressing pregnancy disorders like RSA and preeclampsia. The manuscript is effectively written, engaging, and easy to comprehend. Overall, the experimental design, methodology, statistical analysis, and presentation of results provide compelling support. Nevertheless, there are concerns regarding certain aspects of the manuscript, which are elaborated below: Was there any impact on the cell cycle/polyploidization due to the inhibition of DOCK1 or TBOPP? Figure 3: Kindly verify the expression of DOCK1 in cytosolic and nuclear fractions of HTR8 and JAR cells.

Response: We express our sincere gratitude for your thoughtful review and constructive comments on our manuscript.

Addressing your concerns:

1. Impact on the Cell Cycle/Polyploidization Due to the Inhibition of DOCK1 or TBOPP:

We appreciate your inquiry regarding the impact on the cell cycle and polyploidization due to the inhibition of DOCK1 or TBOPP. To answer your question, we employed flow cytometry to analyze the cell cycle distribution of treated cells. Cells were stained with propidium iodide (PI) post-treatment and were subsequently analyzed using flow cytometry to determine the proportion of cells in different phases of the cell cycle. In our experiments with HTR-8 and JAR cells, we observed that the inhibition of DOCK1 or TBOPP induced cell cycle arrest at the G0/G1 phase, and a concurrent reduction in the S phase was observed. The relationship between cell proliferation and the cell cycle is intrinsic. This cycle begins with the G0/G1 phase, where the cell prepares for DNA synthesis, continues through the S phase, where DNA synthesis actually occurs, and finally into the G2 and M phases, which culminate in cell division. Any interruption or arrest, such as what we observed in the G0/G1 phase due to DOCK1 inhibition, can have effect on the overall proliferative capacity of the cell. The inhibition of DOCK1 exerts a multifaceted impact. Not only does it attenuate the proliferation capacity of trophoblast cells by directly impeding their growth, but it also disrupts the natural progression of the cell cycle, especially stalling it in the G0/G1 phase. We have incorporated this additional data and a comprehensive discussion in the manuscript for your pertinent question. Kindly refer to Fig 2D-E and Fig S2D-E the accompanying text on Page 6, Line 142-152 for the added information.

Fig 2D-E. DOCK1 modulates cell cycle progression in HTR-8 cells. (D) Flow cytometry analysis showing the distribution of HTR-8 cells among G0/G1-, S- and G2/M-phases after DOCK1 knockout (Left), and percentage analysis of cells in each phase relative to the total (Right). (E) After HTR-8 cells were treated with DOCK1 inhibitor 10 μ M TBOPP for 24 h, cell cycle distribution was analyzed by flow cytometry (Left). The respective percentages of cells in each phase are presented (Right).

Fig S2D-E. DOCK1 modulates cell cycle progression in JAR cells. (D) Flow cytometry analysis showing cell cycle distribution for JAR cells after DOCK1 inhibition (Left) and the

relative percentage analysis (Right). (E) JAR cells were treated with 10 μ M TBOPP for 24 h. The distribution of cell cycle was analyzed by flow cytometry (Left). The relative percentages of cells in each phase are depicted (Right).

2. Figure 3: Kindly verify the expression of DOCK1 in cytosolic and nuclear fractions of HTR8 and JAR cells.

Response: Utilizing the nuclear-cytosolic fractionation kit, we assessed DOCK1 protein levels in both HTR-8 and JAR cells. Our findings indicated that DOCK1 was expressed in both the cytosolic and nuclear fractions. In the knockout or inhibition groups, DOCK1 expression was nearly undetectable in both the nuclear and cytosolic fractions. We have made corresponding amendments to Figure 4A and 4B. (Due to the inclusion of additional images related to the cell cycle experiment, we made slight adjustments to the figure arrangement, adding a new figure. As a result, the figure originally referred to by the reviewer as Figure 3 has been updated to Figure 4.)

We hope that these revisions adequately address your comments and facilitate the manuscript's progression to the next stage. Thank you for your suggestions again.

Reviewer #2 (Comments to the Authors (Required)):

This is a well-written manuscript and appropriately designed study looking at the role of DOCK1 pathway associated with RSA. Authors found DOCK1 is reduced at the trophoblast compartment of the RSA placenta, therefore, used HTR8 cell line to characterize cellular functions. Authors derived HTR8 lines with overexpressed and knockdown of DOCK1, and characterized cell proliferation and invasion and found DOCK1 overexpression increases degradation of the DUSP through ubiquitin-proteasome system, leading to inactivation of ERK signaling pathways. Authors also performed in vivo experiments using DOCK1 inhibitor TBOPP, and found that mice treated with TBOPP showed increased rate of embryo

resorption compared to the control. Overall, the manuscript is well-written. I have very little comments for this manuscript summarized below.

1) Figure 6C is difficult to see. DAPI is faint and cannot tell which are nuclear and which are cytoplasmic staining.

Response: We sincerely thank you for your dedicated time and constructive feedback on our manuscript.

Regarding the issue with Figure 6C, we appreciate your keen observation. We have revised the figure to enhance the clarity of DAPI staining and ensure distinct differentiation between nuclear and cytoplasmic staining. We have made corresponding amendments to Figure 7C (Additionally, in response to suggestions from Reviewer 1, we have incorporated additional experimental data, which led to the inclusion of new figures. Consequently, the figure that was initially labeled as Figure 6 has been renumbered to Figure 7. We appreciate your understanding and will ensure clear referencing in the revised manuscript.).

Figure 7C. Interaction of DOCK1 and DUSP4 in the cytoplasm observed by immunofluorescence staining with anti-DOCK1 (red) and anti-DUSP4 (green) in HTR-8 cells.

2) In vivo mouse experiments is very exciting, however, authors only examined the protein expression of key markers, DUSP4, HUWE1, ERK etc, that authors have studied across the manuscript. The issues I have is that whether the treatment of DOCK1 inhibitor is mimicking the RSA pathophysiology or just killing the cells and led to embryo resorption. Just from this experiment, authors cannot rule out/draw the conclusion whether the drugs have the effect on the embryo development, or the actual trophoblastic abnormalities led to the abortion. Authors should consider looking into other cellular phenotypes such as cell proliferation and cell invasion in the mouse placenta to confirm the cellular phenotype is also seen in the TBOPP cases.

Response: RSA involves multiple factors and diverse cell components at the maternal-fetal interface. Several factors, such as infectious, chromosomal, and anatomical factors, endocrine factors, have been implicated in RSA (---Obstet Gynecol Surv. 2022 Jun;77(6):355-366).

Furthermore, aspects like placental defects in proliferation, invasion, migration, maternal immune factors, or excessive apoptosis have also been identified as contributors to its pathogenesis (---Cell Death Dis. 2017 Jun 29;8(6):e2908; ---Hum Reprod Update. 2020 Jun 18;26(4):501-513; ---Genomics Proteomics Bioinformatics. 2021 Apr;19(2):208-222). Our study specifically explored the role of DOCK1 in placental cell proliferation and invasion/migration. The novel DOCK1 inhibitor, TBOPP, has been primarily shown to dampen DOCK1-mediated cell invasion and survival (---Cell Rep. 2017 May 2;19(5):969-980). The aberration in these processes, driven by alterations in DOCK1 function, could be a potential contributing factor to RSA.

We appreciate your suggestion regarding the examination of other cellular phenotypes such as cell invasion and cell proliferation in the mouse placenta to verify the cellular phenotype in the TBOPP cases.

In light of the essential role of trophoblast cell invasion and proliferation in ensuring successful murine pregnancy outcomes, we examined the potential influence of TBOPP on these processes. By employing CK7 (a marker for trophoblastic cells) immunofluorescence staining in the placenta to observe the region of trophoblast invasion into the decidua. Our results showed that the extent of trophoblast invasion into the decidua was noticeably reduced in the TBOPP treatment group (Figure 9F and G). This data supports that the observed embryonic loss under TBOPP treatment might be linked with hindered trophoblast invasiveness. Furthermore, acknowledging the pivotal nature of trophoblast cell proliferation, we next performed immunostaining for Ki67 in placental tissues and found its predominant staining in the labyrinth layer. To analyze the proliferation status, we compared the Ki67 positivity between the TBOPP-treated and control groups. Remarkably, the TBOPP treatment group exhibited a significantly lower percentage of Ki67 positive cells compared to the control group (Figure 9H). These results demonstrate that the DOCK1 inhibitor TBOPP does exert specific effects on trophoblastic cell proliferation and invasion, which could then subsequently influence murine pregnancy outcomes. These additional findings have been integrated into the revised manuscript (see Figure 9F-H, Page 11, Lines 281-294).

Figure 9. DOCK1 deficiency triggers embryo miscarriage in pregnant mice models. (F) Representative HE stained images of placentas and CK7 immunofluorescence of trophoblast cell invasion into the decidua following TBOPP treatment (Left). Based on the HE and CK7 staining, distinct layers - Dec (decidua), Jz (junctional zone) and Lab (labyrinth) – are delineated by black lines in HE images and white lines in immunofluorescence images. Boxes on the left correspond to magnified sections on the right, with the region between the yellow and white lines indicating the trophoblast cell invasion area into the decidua. Scale bar: 500 μ m. (G) The area of trophoblast cell invasion into the decidua was quantified. (H) Representative images of Ki67 immunostaining in placental tissues after TBOPP treatment (Left). Percentage of Ki67-positive cells was calculated (Right). Scale bar: 200 μ m.

Our research has primarily focused on the effects of TBOPP on placental functions, revealing notable changes in trophoblast proliferation and invasion/migration when exposed to this inhibitor. Given the essential role of the placenta in nutrient, gas exchange, and waste removal for the fetus, any disruption in its proper function could contribute to embryonic maldevelopment or loss. Our findings provide insights into the adverse pregnancy outcomes resulting from placental dysfunctions due to TBOPP. However, it's crucial to underscore a limitation in our present study. While our findings shed light on how TBOPP-induced placental dysfunctions might lead to embryonic loss, they don't involve in the direct interactions of TBOPP with embryonic cells. Our forthcoming studies aim to closely examine the direct interplay between TBOPP and embryonic cells, ensuring a holistic understanding of TBOPP's role within the broader landscape of fetal development. We have added relevant content in the discussion section (Page 14, Lines 370-378).

We thank reviewer 2 for above comments that have helped us to improve the clarity and strengthen the conclusions of our manuscript. Thank you for your suggestions again.

November 3, 2023

RE: Life Science Alliance Manuscript #LSA-2023-02247-TR

Prof. Yi Lin
Shanghai Jiao Tong University School of Medicine Affiliated Sixth People's Hospital
Reproductive Medicine Center
No. 600 Yishan Road, Xuhui District, Shanghai, China
Shanghai 200030
China

Dear Dr. Lin,

Thank you for submitting your revised manuscript entitled "DOCK1 insufficiency disrupts trophoblast function and pregnancy disorders via DUSP4-ERK pathway". We would be happy to publish your paper in Life Science Alliance pending final revisions necessary to meet our formatting guidelines.

- please add ORCID ID for the corresponding author--you should have received instructions on how to do so
- please add the Twitter handle of your host institute/organization as well as your own or/and one of the authors in our system
- you've indicated activity for Surendra Sharma that does not qualify a contributor for authorship. Please either update this activity, or let us know if the author should be removed.

A. FINAL FILES:

B. MANUSCRIPT ORGANIZATION AND FORMATTING:

Sincerely,

November 7, 2023

RE: Life Science Alliance Manuscript #LSA-2023-02247-TRR

Prof. Yi Lin
Shanghai Jiao Tong University School of Medicine Affiliated Sixth People's Hospital
Reproductive Medicine Center
No. 600 Yishan Road, Xuhui District
Shanghai 200030
China

Dear Dr. Lin,

Thank you for submitting your Research Article entitled "DOCK1 insufficiency disrupts trophoblast function and pregnancy disorders via DUSP4-ERK pathway". It is a pleasure to let you know that your manuscript is now accepted for publication in Life Science Alliance. Congratulations on this interesting work.

DISTRIBUTION OF MATERIALS:

Again, congratulations on a very nice paper. I hope you found the review process to be constructive and are pleased with how the manuscript was handled editorially. We look forward to future exciting submissions from your lab.

Sincerely,
